# *SoTeacher*: Toward Student-oriented Teacher Network Training for Knowledge Distillation

## Abstract

How to train an ideal teacher for knowledge distillation is still an open problem. It has been widely observed that a best-performing teacher does not necessarily yield the best-performing student, suggesting a fundamental discrepancy between the current practice in teacher training and the distillation objective. To fill this gap, we explore the feasibility of training a teacher that is oriented toward student performance with empirical risk minimization. Our analyses are inspired by the recent findings that the effectiveness of knowledge distillation hinges on the teacher's capability to approximate the true label distribution of training inputs. We theoretically established that (1) the empirical risk minimizer can provably approximate the true label distribution of training data if the loss function is a proper scoring rule and the hypothesis function is locally-Lipschitz continuous around training inputs; and (2) when data augmentation is employed for training, an additional constraint is required that the minimizer has to produce consistent predictions across augmented views of the same training input. In light of our theory, we propose a teacher training method *SoTeacher* which renovates the empirical risk minimization by incorporating Lipschitz regularization and consistency regularization. Experiments on two benchmark datasets confirm that *SoTeacher* can improve student performance significantly and consistently across various knowledge distillation algorithms and teacher-student pairs.

## 1 Introduction

Knowledge distillation aims to train a small yet effective *student* neural network following the guidance of a large *teacher* neural network (Hinton et al., 2015). It dates back to the pioneering idea of model compression (Buciluă et al., 2006) and has a wide spectrum of real-world applications, such as recommender systems (Tang & Wang, 2018; Zhang et al., 2020), question answering systems (Yang et al., 2020; Wang et al., 2020) and machine translation (Liu et al., 2020).

Despite the prosperous research interests in knowledge distillation, one of its crucial components, teacher training, is largely neglected. Existing training practice of teacher networks is often directly targeting at maximizing the performance of the teacher, which does not necessarily transfer to the performance of the student. Empirical evidence shows that a teacher trained toward convergence will yield an inferior student (Cho & Hariharan, 2019) and regularization methods benefitting the teacher may contradictorily degrade student performance (Müller et al., 2019). As also shown in Figure 1, the teacher trained towards convergence will consistently reduce the performance of the student after a certain point. This suggests a fundamental discrepancy between the common practice in neural network training and the learning objective of knowledge distillation.

In this work, we explore both the theoretical feasibility and practical methodology of training the teacher toward student performance. Our analyses are built upon the recent understanding of knowledge distillation from a statistical perspective. In specific, Menon et al. (2021) show that the soft prediction provided by the teacher is essentially an approximation to the true label distribution, and true label distribution as supervision for the student improves the generalization bound compared to one-hot labels. Dao et al. (2021) show that the accuracy of the student is directly bounded by the distance between teacher's prediction and the true label distribution through the Rademacher analysis.

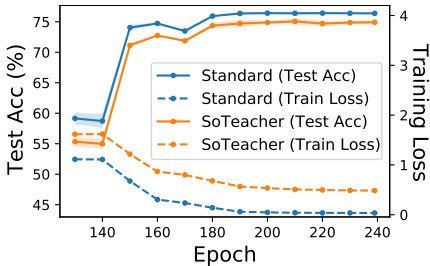 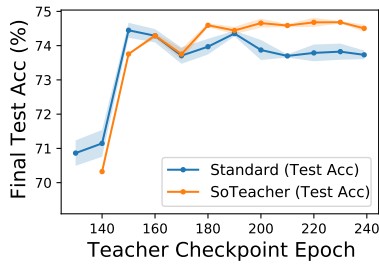

(a) Test accuracy and training loss of the teacher at different epochs in training.

(b) Student performance when distilled from different teacher checkpoints.

Figure 1: On CIFAR-100, we store a teacher checkpoint every 10 epochs and distill a student from it. We can observe that: (1) a standard teacher trained towards better teacher performance may consistently deteriorate student performance upon knowledge distillation; (2) a teacher trained using *SoTeacher* can achieve better student performance even with lower teacher performance.

Based on the above understanding, a teacher benefitting the student should be able to learn the true label distribution of the *distillation data* [1]. Since practically the distillation data is often reused from the teacher's training data, the teacher will have to learn the true label distribution of its own training data. This might appear to be infeasible using standard empirical risk minimization, as the teacher network often has enough capacity to fit all one-hot training labels, in which case, distilling from teacher predictions should not outperform direct training based on one-hot labels. Previous theoretical analyses tend to evade this dilemma by distilling from teacher predictions only on data that is not used in teacher training (Menon et al., 2021; Dao et al., 2021).

Instead, we directly prove the feasibility of training the teacher to learn the true label distribution of its training data. We show that the standard empirical risk minimizer can approach the true label distribution of training data as long as the loss function is a proper scoring rule and the hypothesis function is locally Lipschitz continuous around training samples. We further show that when data augmentation is employed for training, our argument still holds true under an additional constraint, *i.e.*, predictions on the same training input under different augmentations have to be consistent.

In light of our theory, we show that explicitly imposing the Lipschitz and consistency constraint in teacher training can facilitate the learning of the true label distribution and thus improve the student performance. We conduct extensive experiments on two benchmark datasets using various knowledge distillation algorithms and different teacher-student architecture pairs. The results confirm that our method can improve student performance consistently and significantly.

To summarize, our main contributions can be listed as follows.

- We show that it is theoretically feasible to train the teacher to learn the true label distribution of the distillation data even with data reuse, which explains why the current knowledge distillation practice works.
- We show that explicitly imposing the Lipschitz and consistency regularization in teacher training can better learn the true label distribution and improve the effectiveness of knowledge distillation.

We believe our work is among the first attempts to explore the theory and practice of training a *student-oriented* teacher in knowledge distillation. We hope our exploration can serve as a stepping stone to rethinking the teacher training and unleashing the full potential of knowledge distillation.

## 2 PROBLEM FORMULATION

**Standard practice of teacher network training.** We study knowledge distillation in the context of multi-class classification. Specifically, we are given a set of training samples $\mathcal{D} = \{(x_i, y_i)\}_{i \in [N]}$, where $[N] \equiv 1, 2, \cdots, N$. $\mathcal{D}$ is drawn from a probability distribution $p_{X,Y}$ that is defined jointly over input space $\mathcal{X} \subset \mathbb{R}^d$ and label space $\mathcal{Y} = [K]$. In common practice, the teacher network

---

[1]For simplicity, we refer to the student's training data in knowledge distillation as the distillation data (Stanton et al., 2021)

$\boldsymbol{f} : \mathcal{X} \to [0, 1]^{|\mathcal{Y}|}$ is trained to minimize the empirical risk given a loss function $\ell$, namely

$$\boldsymbol{f}^* = \arg \min_{\boldsymbol{f}} \mathbb{E}_{(x,y) \in \mathcal{D}} \, \ell(\boldsymbol{f}(x), y), \tag{1}$$

where $\mathbb{E}_{(x,y) \in \mathcal{D}}$ is the empirical expectation.

**Learning objective of teacher in knowledge distillation.** Recent advances in understanding knowledge distillation suggest a teacher should approximate the true label distribution of the distillation data, which is often reused from the teacher's training data. This implies

$$\boldsymbol{f}^* = \arg \min_{\boldsymbol{f}} \mathbb{E}_{(x,y) \in \mathcal{D}} \, \|\boldsymbol{f}(x) - \boldsymbol{p}^*(x)\|. \tag{2}$$

Here, $\boldsymbol{p}^*(x) = p_{X,Y}(\cdot|x)$ denotes the *true label distribution* of an input $x$, namely the (unknown) category distribution that its label is sampled from, which is not necessarily one-hot. And $\|\cdot\|$ is a distance function induced by an arbitrary $p$-norm.

**Our Research Questions.** One can find that there is a fundamental discrepancy between the common practice of teacher training (Eq. (1)) and the learning objective of the teacher in knowledge distillation (Eq. (2)). In particular, if the minimizer $\boldsymbol{f}^*$ in Eq. (1) is attained, $\boldsymbol{f}^*(x) = \boldsymbol{1}_y$ for any distillation input $x$, which significantly deviates from $\boldsymbol{p}^*(x)$ and challenges the effectiveness of knowledge distillation. Therefore, in this work, we explore the following two questions.

(i) *Can a teacher network learn the true label distribution of the training data with the standard teacher training practice?*
(ii) *How to train a teacher to better learn the true label distribution and improve student performance?*

We will present our findings of these two questions in Sec. 3 and Sec. 4, respectively.

## 3 THEORETICAL FEASIBILITY TO LEARN TRUE LABEL DISTRIBUTION OF TRAINING DATA

We now explore the theoretical feasibility of training a teacher that learns the true label distribution of the training data under the empirical risk minimization framework. Note that throughout the discussion here, our major focus is the *existence* of a proper minimizer, instead of the details of the optimization process.

### 3.1 A HYPOTHETICAL CASE WITH IDENTIFIABLE SIBLINGS

We first show that in a hypothetical case where the siblings of an input are known, the empirical minimizer can produce predictions close to the true label distribution on the training data. We define the notion of siblings as the set of all inputs sharing the same true label distribution as $x$ in the set of training inputs $\mathcal{D}_{\mathcal{X}} = \{x | (x, y) \in \mathcal{D}\}$.

**Definition 3.1** (Siblings). *We define the siblings of $x$ in $\mathcal{D}_{\mathcal{X}}$ under $\boldsymbol{p}^*$ as $\mathcal{S}_{\boldsymbol{p}^*}(x) = \{x' \in \mathcal{D}_{\mathcal{X}} | \boldsymbol{p}^*(x') = \boldsymbol{p}^*(x)\}$.*

The siblings of $x$ have the property that the sample mean of their labels converges to $x$'s true label distribution, namely $\mathbb{E}_{x \in \mathcal{S}_{\boldsymbol{p}^*}(x)}[\boldsymbol{1}_y] \to \boldsymbol{p}^*(x)$. To see that, note the label $y$ of an input $x$ is a random variable following the true label distribution, namely $y|x \sim p_{Y|X}(y|x)$ and thus $\mathbb{E}[\boldsymbol{1}_y] = \boldsymbol{p}^*(x)$ (Neal, 2007). And per the definition of siblings, $\boldsymbol{p}^*(x') = \boldsymbol{p}^*(x)$ for all inputs $x' \in \mathcal{S}_{\boldsymbol{p}^*}(x)$, which means that the labels of the siblings are identically distributed.

We now propose that if the loss function is properly selected and the network function is properly regularized, the network minimizes the empirical risk on $\mathcal{S}_{\boldsymbol{p}^*}(x)$ will produce predictions close to the true label distribution. We first introduce the notion of a *proper scoring rule* (Gneiting & Raftery, 2007) and then discuss this proposition.

**Definition 3.2** (Proper scoring rule). *Let $\mathcal{E}$ be a class of probability mass functions defined on $\mathcal{Y}$. $\ell(\cdot, \cdot)$ is a proper scoring rule with respect to $\mathcal{E}$ if for all $\boldsymbol{p}, \boldsymbol{q} \in \mathcal{E}$, $\ell(\boldsymbol{p}, \boldsymbol{q}) \geq \ell(\boldsymbol{q}, \boldsymbol{q})$, with equality if and only if $\boldsymbol{p} = \boldsymbol{q}$. Note that many common loss functions are proper scoring rules (Lakshminarayanan et al., 2017), including the negative-log likelihood loss (NLL) and the Mean Square Error (MSE) (also known as the Brier score (Brier, 1950)).*

**Proposition 3.3** (Approximation error of the empirical risk minimizer on siblings). *Let $\ell(\cdot, \cdot)$ be a proper scoring rule. Let $\mathcal{F}$ be a family of functions that attains the same prediction on the siblings of $x$, namely for all $\boldsymbol{f} \in \mathcal{F}$, $\boldsymbol{f}(x') = \boldsymbol{f}(x)$, $\forall x' \in \mathcal{S}_{\boldsymbol{p}^*}(x)$. Let $\boldsymbol{f}^* \in \mathcal{F}$ be a function that minimizes the empirical risk on $\mathcal{S}_{\boldsymbol{p}^*}(x)$. Then for any $x \in \mathcal{S}_{\boldsymbol{p}^*}(x)$, with probability at least $1 - \delta$,*

$$\|\boldsymbol{f}^*(x) - \boldsymbol{p}^*(x)\| \leq \sqrt{\frac{2K}{|\mathcal{S}_{\boldsymbol{p}^*}(x)|} \log \frac{2}{\delta}}. \tag{3}$$

The proof is rather straightforward, as based on the property of the proper scoring rule, $\boldsymbol{f}$ can only achieve the minimum when $\boldsymbol{f}(x) = \mathbb{E}_{x \in S_{\boldsymbol{p}^*}(x)}[\mathbf{1}_y]$, which converges to the true label distribution as mentioned above. For example, when $\ell$ is NLL, we have $\hat{\mathbb{E}}[-\mathbf{1}_y \cdot \log \boldsymbol{f}(x)] = -\hat{\mathbb{E}}[\mathbf{1}_y] \cdot \log \boldsymbol{f}(x) \geq -\hat{\mathbb{E}}[\mathbf{1}_y] \cdot \log \hat{\mathbb{E}}[\mathbf{1}_y]$, with the minimum attained if and only if $\boldsymbol{f}(x) = \hat{\mathbb{E}}[\mathbf{1}_y]$ (the sample mean is abbreviated as $\hat{\mathbb{E}}$ for simplicity). Note that the last inequality here is in fact Gibbs' inequality.

### 3.2 REALISTIC CASE

Proposition 3.3 requires identifying the siblings of an input $x$ in $\mathcal{D}_\mathcal{X}$ under the true label distribution $\boldsymbol{p}^*$, which might not possible in a realistic scenario, because $\boldsymbol{p}^*$ is usually unknown and $\mathcal{D}_\mathcal{X}$ may not necessarily contain the siblings of $x$. We now introduce the notion of neighborhood and show such a requirement can be relaxed.

**Definition 3.4** (Neighborhood). *The neighborhood of an input $x$ is defined as the set of inputs that are close to $x$, namely $\mathcal{N}(x) = \{x' \in \mathcal{D}_\mathcal{X} | \|x' - x\| \leq r\}$.*

Now if we assume the true label distribution $\boldsymbol{p}^*$ is $L$-locally-Lipschitz continuous, we will have for any $x' \in \mathcal{N}(x)$, $\|\boldsymbol{p}^*(x') - \boldsymbol{p}^*(x)\| \leq Lr$. This means that any input in the neighborhood of $x$ shares a similar true label distribution as $x$, which is a relaxation of its siblings with an error controlled by $L$.

On the neighborhood set of $x$, we can still show that the empirical risk minimizer will produce predictions close to the true label distribution. However, due to the relaxation, additional error terms will be introduced to the approximation bound Eq. (3). And similar to siblings, the constraints on the loss function and network function would still be necessary. First, the loss function has to be a proper scoring rule. Second and slightly differently, the network function has to be locally-Lipschitz continuous in the neighborhood of $x$.

Finally, to establish the approximation bound on all training inputs, we need to partition the training set into multiple neighborhoods where the constraint of the empirical risk minimizer can be satisfied individually. We achieve this by introducing a *covering*.

**Definition 3.5** (r-external covering). *We say $\mathcal{C}$ is a r-external covering of $\mathcal{D}_\mathcal{X}$, if $\mathcal{D}_\mathcal{X} \subseteq \bigcup_{x \in \mathcal{C}} \{x' \in \mathcal{X} \mid \|x' - x\| \leq r\}$. We denote $N^r$ as the covering number, namely the minimum cardinality of any external covering of $\mathcal{D}_\mathcal{X}$.*

We are now ready to present our main theorem.

**Theorem 3.6** (Approximation error of the empirical risk minimizer to the true label distribution ). *Assume $\boldsymbol{p}^*$ is $L^*$-locally-Lipschitz continuous in a norm ball of radius $r$ around $x \in \mathcal{X}$. Let $\ell(\cdot, \cdot)$ be a proper scoring rule. Let $\mathcal{C}$ be an $r$-external covering of $\mathcal{D}_\mathcal{X}$. Let $\mathcal{F}$ be a family of functions that are $L$-locally-Lipschitz continuous in a norm ball of radius $r$ around $x \in \mathcal{C}$. Let $\boldsymbol{f}^* \in \mathcal{F}$ be a function that minimizes the empirical risk Eq. (1). Let $\kappa \geq 1$ be a number and $\hat{\mathcal{D}}_\mathcal{X}$ be a subset of $\mathcal{D}_\mathcal{X}$ with cardinality at least $(1 - 1/\kappa + 1/(\kappa N^r))N$. Then for any $x \in \tilde{\mathcal{D}}_\mathcal{X}$, any p-norm $\| \cdot \|$, with probability at least $1 - \delta$, we have*

$$\|\boldsymbol{f}^*(x) - \boldsymbol{p}^*(x)\| \leq \sqrt{\frac{2\kappa N^r K}{N} \log \frac{2}{\delta}} + Lr\xi_K + 2L^*r. \tag{4}$$

*Here $\xi_K = (1 + K\|\mathbf{1}_{(\cdot)} - K^{-1}\mathbf{1}\|)$, where $\mathbf{1}_{(\cdot)} \in \mathbb{R}^K$ denotes the vector where an arbitrary entry is 1 and others are 0, and $\mathbf{1} \in \mathbb{R}^K$ denotes the vector where all entries are 1.*

One can find that as long as $L^*$ and $L$ are small enough, and the size of the training set $N$ is sufficiently large, the predictions can approach the true label distribution on a majority of the training inputs with

high probability. This suggests that a teacher trained with standard empirical risk minimization is still possible to learn the true label distribution of its training data, thus resolving the dilemma of why knowledge distillation with data reuse can improve student generalization. Eq. (4) also shows that the bound will be tighter if $K$ is small, which implies learning the true label distribution of the training data will be significantly easier on classification tasks with fewer classes.

### 3.3 THE CASE WITH DATA AUGMENTATION

When data augmentation is employed in training, we note that the neighborhood defined in Sect. 3.2 needs to be relaxed to include augmented training inputs. We now define the augmented neighborhood.

**Definition 3.7** (Augmented neighborhood). *Let $\mathcal{A} : \mathcal{X} \to \mathcal{X}$ denote a set of data augmentation functions. The augmented neighborhood of an input $x$ is defined as the set of inputs that are close to any augmented $x$, namely $\mathcal{N}_\mathcal{A}(x) = \{x' \in \mathcal{D}_\mathcal{X} | \, \|x' - A(x)\| \le r$, for any $A \in \mathcal{A}\}$.*

Now if we assume the true label distribution is nearly invariant under data augmentations, namely $\|\boldsymbol{p}^*(A(x)) - \boldsymbol{p}^*(x)\| \le L'$, we will still have for any $x' \in \mathcal{N}_\mathcal{A}(x)$, $\|\boldsymbol{p}^*(x') - \boldsymbol{p}^*(x)\| \le Lr + L'$. Therefore, the approximation bound on the neighborhood can be readily extended to the augmented neighborhood here.

**Theorem 3.8** (Approximation error of the empirical loss minimizer with data augmentation). *Consider the same setting as Theorem 3.6, except each training input is now augmented by a set of augmentation functions $\mathcal{A}$. The empirical risk minimizer now becomes*

$$\boldsymbol{f}^* = \arg \min_{\boldsymbol{f} \in \mathcal{F}} \mathbb{E}_{(x,y) \in \mathcal{D}} \mathbb{E}_{A \in \mathcal{A}} \, \ell(\boldsymbol{f}(A(x)), y). \tag{5}$$

*Then with probability at least $1 - \delta$, for any $x \in \tilde{\mathcal{D}}_\mathcal{X}$,*

$$\|\boldsymbol{f}^*(x) - \boldsymbol{p}^*(x)\| \le \sqrt{\frac{2\kappa N^r K}{N} \log \frac{2}{\delta}} + Lr\xi_K + 2L^*r + L_f^A \xi_K + L_p^A, \tag{6}$$

*where $L_f^A = \max_{x \in \mathcal{C}, A \in \mathcal{A}, x' \in \mathcal{N}_\mathcal{A}(x)} \|\boldsymbol{f}^*(x') - \boldsymbol{f}^*(A(x))\|$, $L_p^A = \max_{x \in \mathcal{C}, A \in \mathcal{A}, x' \in \mathcal{N}_\mathcal{A}(x)} \|\boldsymbol{p}^*(x') - \boldsymbol{p}^*(A(x))\|$.*

Here $L_f^A$ and $L_p^A$ cannot be simply bounded by the local Lipschitz continuities of the hypothesis function and true label distribution respectively, because the augmentation function $A$ can significantly change the data vector of the input, which means $\|A(x) - x\|$ is not necessarily small. Therefore, to ensure the true label distribution is properly learned with data augmentation, one has to explicitly minimize these two error terms.

Note that although the training set is larger after data augmentation, the bound Eq. (6) is not improved by a multiplier compared to the case with no augmentation Eq. (4). This is because the labels of augmented inputs are statistically dependent, thus the bound in the concentration inequality will not become tighter (see Lemma A.2).

## 4 PRACTICAL METHOD TO IMPROVE THE LEARNING OF TRUE LABEL DISTRIBUTION OF THE TRAINING DATA

Following our theory, we explore practical methods to train the teacher to better learn the true label distribution of the training data.

**Loss function.**     Theorem 3.6 suggests that the necessary condition to learn the true label distribution is to select a proper scoring rule as the loss function. This is already sufficed by current teacher training practice where the loss function is typically NLL or MSE.

**Lipschitz regularization.**     Theorem 3.6 also suggests that it is necessary to enforce the hypothesis function to be locally Lipschitz continuous around certain samples. This may also be achieved by current teacher training practice, as neural networks are often implicitly Lipschitz bounded (Bartlett et al., 2017). Nevertheless, we observe that explicit Lipschitz regularization (LR) can still help in multiple experiments (See Sect. 5). Therefore, we propose to incorporate a global Lipschitz constraint into teacher's training. For the implementation details, we follow the existing practice of Lipschitz regularization (Yoshida & Miyato, 2017; Miyato et al., 2018) and defer them to Appendix C.

**Consistency regularization.** When data augmentation is employed in training, Theorem 3.8 suggests that it is necessary to ensure both the true label distribution and the teacher prediction of an input will not change significantly after data augmentations, We note that these two constraints are missing in the current teacher training practice and thus have to be explicitly imposed. The former can be sufficed by selecting proper data augmentations such as random cropping, rotation or flipping commonly seen in the training practice. The latter is known as the consistency regularization (CR) (Laine & Aila, 2017; Xie et al., 2020; Berthelot et al., 2019; Sohn et al., 2020) that is widely used in semi-supervised learning.

To maximize the training efficiency of consistency regularization, we utilize temporal ensembling (Laine & Aila, 2017), which penalizes the difference between the current prediction and the aggregation of previous predictions for each training input. In this way, the consistency under data augmentation is implicitly regularized since the augmentation is randomly sampled in each epoch. And it is also efficient as no extra model evaluation is required for an input.

To scale the consistency loss, a loss weight is often adjusted based on a Gaussian ramp-up curve in previous works (Laine & Aila, 2017). However, the specific parameterization of such a Gaussian curve varies greatly across different implementations, where more than one additional hyperparameters have to be set up and tuned heuristically. Here to avoid tedious hyperparameter tuning we simply linearly interpolate the weight from 0 to its maximum value, namely $\lambda_{\mathrm{CR}}(t) = \frac{t}{T}\lambda_{\mathrm{CR}}^{\max}$, where $T$ is the total number of epochs.

**Summary.** To recap, our teacher training method introduces two additional regularization terms. The loss function can thus be defined as $\ell = \ell_{\mathrm{Stand.}} + \lambda_{\mathrm{LR}}\ell_{\mathrm{LR}} + \lambda_{\mathrm{CR}}\ell_{\mathrm{CR}}$, where $\ell_{\mathrm{Stand.}}$ is the standard empirical risk defined in Eq.(1) and $\lambda_{\mathrm{LR}}$ is the weight for Lipschitz regularization. Our method is simple to implement and incurs only minimal computation overhead (see Section 5.2).

## 5 EXPERIMENTS

In this section, we evaluate the effectiveness of our teacher training method in knowledge distillation. We focus on compressing a large network to a smaller one where the student is trained on the same data set as the teacher, which is well-supported by our theoretical understanding.

### 5.1 EXPERIMENT SETUP

We denote our method as student-oriented teacher training (*SoTeacher*), since it aims to learn the true label distribution to improve student performance, rather than to maximize teacher performance. We compare our method with the standard practice for teacher training in knowledge distillation (*Standard*) (*i.e.*, Eq. (1)). We conduct experiments on benchmark datasets including CIFAR-100 (Krizhevsky, 2009) and Tiny-ImageNet (Tin, 2017), a miniature version of ImageNet (Deng et al., 2009) consisting of $100,000$ training samples selected from $200$ classes. Unfortunately, due to computational constraints, we were not able to conduct experiments on ImageNet. We experiment with various backbone networks including ResNet (He et al., 2016), Wide ResNet (Zagoruyko & Komodakis, 2016) and ShuffleNet (Zhang et al., 2018b; Tan et al., 2019). We test the applicability of *SoTeacher* from different aspects of model compression in knowledge distillation including reduction of the width or depth, and distillation between heterogeneous neural architectures.

For knowledge distillation algorithms, we experiment with the original knowledge distillation method (KD) (Hinton et al., 2015), and a wide variety of other sophisticated knowledge distillation algorithms (see Sect. 5.2). We report the classification accuracies on the test set of both teacher and the student distilled from it. All results are presented with mean and standard deviation based on 3 independent runs. For Tiny-ImageNet, we also report the top-5 accuracy. For hyperparameters, we set $\lambda_{\mathrm{CR}}^{\max} = 1$ for both datasets. For Lipschitz regularization, we set $\lambda_{\mathrm{LR}} = 10^{-5}$ for CIFAR-100 and $\lambda_{\mathrm{LR}} = 10^{-6}$ for Tiny-ImageNet, as the regularization term is the sum of the Lipschitz constants of trainable parameters, and the teacher model for Tiny-ImageNet has approximately 10 times more parameters than that for CIFAR-100. More detailed hyperparameter settings for neural network training and knowledge distillation algorithms can be found in Appendix D.

### 5.2 RESULTS

**End-to-end Knowledge Distillation Performance.** Tables 1 and 2 show the evaluation results on CIFAR-100 and Tiny-ImageNet, respectively. Our teacher training method SoTeacher can improve

Table 1: Test accuracy of the teacher and student with knowledge distillation conducted on CIFAR-100. *SoTeacher* achieves better student accuracy than Standard for various architectures, depsite a lower teacher accuracy.

|  | WRN40-2/WRN40-1 | | WRN40-2/WRN16-2 | | ResNet32x4/ShuffleNetV2 | |
|---|---|---|---|---|---|---|
|  | **Student** | Teacher | **Student** | Teacher | **Student** | Teacher |
| Standard | $73.73 \pm 0.13$ | $76.38 \pm 0.13$ | $74.87 \pm 0.45$ | $76.38 \pm 0.13$ | $74.86 \pm 0.18$ | $79.22 \pm 0.03$ |
| *SoTeacher* | $\mathbf{74.35} \pm 0.23$ | $74.95 \pm 0.28$ | $\mathbf{75.39} \pm 0.23$ | $74.95 \pm 0.28$ | $\mathbf{77.24} \pm 0.09$ | $78.49 \pm 0.09$ |
| no-CR | $74.34 \pm 0.11$ | $74.30 \pm 0.12$ | $75.20 \pm 0.24$ | $74.30 \pm 0.12$ | $76.52 \pm 0.52$ | $77.73 \pm 0.17$ |
| no-LR | $73.81 \pm 0.15$ | $76.71 \pm 0.16$ | $75.21 \pm 0.13$ | $76.71 \pm 0.16$ | $76.23 \pm 0.18$ | $80.01 \pm 0.18$ |

Table 2: Test accuracy of the teacher and student with knowledge distillation conducted on Tiny-ImageNet. The teacher network is ResNet34 and the student network is ResNet18. *SoTeacher* achieves better student accuracy than Standard with a lower teacher Top-1 accuracy.

|  | **Student** (Top-1) | **Student** (Top-5) | Teacher (Top-1) | Teacher (Top-5) |
|---|---|---|---|---|
| Standard | $66.19 \pm 0.17$ | $85.74 \pm 0.21$ | $64.94 \pm 0.32$ | $84.33 \pm 0.40$ |
| *SoTeacher* | $\mathbf{66.83} \pm 0.20$ | $86.19 \pm 0.22$ | $64.88 \pm 0.48$ | $84.91 \pm 0.41$ |
| no-CR | $66.39 \pm 0.27$ | $86.05 \pm 0.17$ | $64.36 \pm 0.43$ | $84.10 \pm 0.27$ |
| no-LR | $66.48 \pm 0.43$ | $\mathbf{86.20} \pm 0.40$ | $64.26 \pm 1.54$ | $84.48 \pm 0.84$ |

the student's test accuracy consistently across different datasets and teacher/student architecture pairs. Note that the success of our teacher training method is not due to high accuracy of the teacher. In Tables 1 and 2, one may already notice that our regularization method will hurt the accuracy of the teacher, despite that it can improve the accuracy of the student distilled from it.

Table 3: Estimation of the uncertainty quality of the teacher network trained by Standard and *SoTeacher*. The uncertainty quality is estimated by the ECE and NLL both before and after temperature scaling (TS).

| Dataset | Teacher | Method | ECE | NLL | ECE (w/ TS) | NLL (w/ TS) |
|---|---|---|---|---|---|---|
| CIFAR-100 | WRN40-2 | Standard | $0.113 \pm 0.003$ | $1.047 \pm 0.007$ | $0.028 \pm 0.004$ | $0.905 \pm 0.008$ |
|  |  | *SoTeacher* | $0.057 \pm 0.002$ | $0.911 \pm 0.013$ | $0.016 \pm 0.003$ | $0.876 \pm 0.012$ |
| CIFAR-100 | ResNet32x4 | Standard | $0.083 \pm 0.003$ | $0.871 \pm 0.010$ | $0.036 \pm 0.001$ | $0.815 \pm 0.008$ |
|  |  | *SoTeacher* | $0.037 \pm 0.001$ | $0.777 \pm 0.006$ | $0.021 \pm 0.001$ | $0.764 \pm 0.003$ |
| Tiny-ImageNet | ResNet34 | Standard | $0.107 \pm 0.007$ | $1.601 \pm 0.037$ | $0.043 \pm 0.002$ | $1.574 \pm 0.015$ |
|  |  | *SoTeacher* | $0.070 \pm 0.007$ | $1.496 \pm 0.031$ | $0.028 \pm 0.002$ | $1.505 \pm 0.033$ |

To further interpret the success of our teacher training method, we show that our regularization can indeed improve the approximation of the true label distribution thus benefiting the student generalization. Directly measuring the quality of the true distribution approximation is infeasible as the true distribution is unknown for realistic datasets. Follow previous works (Menon et al., 2021), we instead *estimate* the approximation quality by reporting the Expected Calibration Error (ECE) (Guo et al., 2017) and NLL loss of the teacher on a holdout set with one-hot labels. Since scaling the teacher predictions in knowledge distillation can improve the uncertainty quality (Menon et al., 2021), we also report ECE and NLL after temperature scaling, where the optimal temperature is located on an additional holdout set (Guo et al., 2017). As shown in Table 3, our teacher training method can consistently improve the approximation quality for different datasets and teacher architectures.

**Ablation Study.** We toggle off the Lipschitz regularization (LR) or consistency regularization (CR) in *SoTeacher* (denoted as *no-LR* and *no-CR*, respectively) to explore their individual effects. As shown in Tables 1 and 2, LR and CR can both improve the performance individually. But on average, *SoTeacher* achieves the best performance when combining both LR and CR, as also demonstrated in our theoretical analyses. Note that in Table 2, using Lipschitz regularization is not particularly effective because the regularization weight might not be properly tuned (see Figure 2).

**Effect of Hyperparameters.** We conduct additional experiments on Tiny-ImageNet as an example to study the effect of two regularization terms introduced by our teacher training method. For

Lipschitz regularization, we modulate the regularization weight $\lambda_{\text{LR}}$. For consistency regularization, we try different maximum regularization weights $\lambda_{\text{CR}}^{\max}$ and different weight schedulers including linear, cosine, cyclic, and piecewise curves. Detailed descriptions of these schedulers can be found in Appendix D. As shown in Figure 2, both Lipschitz and consistency regularizations can benefit the teacher training in terms of the student generalization consistently for different hyperparameter settings. This demonstrates that our regularizations are not sensitive to hyperparameter selection. Note that the hyperparameter chosen to report the results in Tables 1 and 2 might not be optimal since we didn't perform extensive hyperparameter search in fear of overfitting small datasets. It is thus possible to further boost the performance by careful hyperparameter tuning.

In particular, Figure 2(a) shows that, as Lipschitz regularization becomes stronger, the teacher accuracy constantly decreases while the student accuracy increases and converges. This demonstrates that excessively strong Lipschitz regularization hurts the performance of neural network training, but it can help student generalization in the knowledge distillation context.

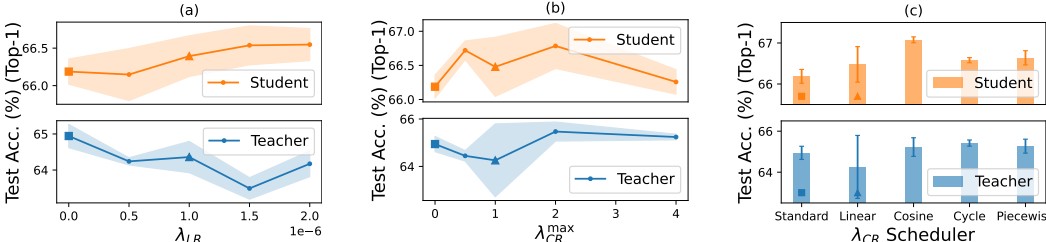

Figure 2: Effect of varying the hyperparameters in our teacher training method, including the weight for Lipschitz regularization $\lambda_{\text{LR}}$, the weight for consistency regularization $\lambda_{\text{CR}}$, and its scheduler. The settings of our method used to report the results (e.g. Table 2) are denoted as "▲". The standard teacher training practice is denoted as "■" for comparison.

**Other Knowledge Distillation Algorithms.** Besides the original knowledge distillation algorithm, we experiment with various feature distillation algorithms including FitNets (Romero et al., 2015), AT (Zagoruyko & Komodakis, 2017), SP (Tung & Mori, 2019), CC (Peng et al., 2019), VID (Ahn et al., 2019), RKD (Park et al., 2019), PKT (Passalis & Tefas, 2018), AB (Heo et al., 2019), FT (Kim et al., 2018), NST (Huang & Wang, 2017), CRD (Tian et al., 2020) and SSKD (Xu et al., 2020). For the implementation of these algorithms, we refer to existing repositories for knowledge distillation (Tian et al., 2020; Shah et al., 2020; Matsubara, 2021) and author-provided codes. Although these distillation algorithms match all types of features instead of predictions between teacher and student, they will achieve the best distillation performance when combined with the prediction distillation (*i.e.* original KD). Therefore, our teacher training method should still benefit the effectiveness of these distillation algorithms. We also experiment with a curriculum distillation algorithm RCO (Jin et al., 2019) which distills from multiple checkpoints in teacher's training trajectory. Our teacher training method should also benefit RCO as the later teacher checkpoints become more student-oriented. As shown in Table 6, our *SoTeacher* can boost the distillation performance of almost all these distillation algorithms, demonstrating its wide applicability.

**Student fidelity.** Recent works have underlined the importance of student fidelity in knowledge distillation, namely the ability of the student to match teacher predictions (Stanton et al., 2021). Student fidelity can be viewed as a measure of knowledge distillation effectiveness that is orthogonal to student generalization, as the student is often unable to match the teacher predictions although its accuracy on unseen data improves (Furlanello et al., 2018; Mobahi et al., 2020). Here we measure the student fidelity by the average agreement between the student and teacher's top-1 predicted labels on the test set. As shown in Table 4, our teacher training method can consistently and significantly improve the student fidelity for different datasets and teacher-student pairs, which aligns with the improvement of student generalization shown by Table 1 and 2. This demonstrates that the teacher can better transfer its "knowledge" to the student with our training method.

**Training overhead.** Compared to standard teacher training, the computation overhead of *SoTeacher* is mainly due to the calculation of the Lipschitz constant, which is efficient as it only requires simple arithmetic calculations of the trainable weights of a neural network (see Section 4). Empirically we observe that training with *SoTeacher* is only slightly longer than the standard training for about $5\%$. The memory overhead of *SoTeacher* is incurred by buffering an average prediction for each

Table 4: Average agreement (%) between the student and teacher's top-1 predictions on the test set.

| | CIFAR-100 | | | Tiny-ImageNet |
| --- | --- | --- | --- | --- |
| | WRN-40-2/WRN-40-1 | WRN-40-2/WRN-16-2 | ResNet32x4/ShuffleNetV2 | ResNet34/ResNet18 |
| Standard | $76.16 \pm 0.14$ | $76.92 \pm 0.29$ | $76.63 \pm 0.25$ | $71.33 \pm 0.07$ |
| *SoTeacher* | $77.92 \pm 0.27$ | $79.41 \pm 0.11$ | $80.36 \pm 0.13$ | $73.36 \pm 0.25$ |

input. However, since such prediction requires no gradient calculation we can simply store it in a memory-mapped file.

## 6 RELATED WORK

**Alleviate "teacher overfitting".** Since in knowledge distillation the distillation data is often reused from the teacher's training data, a teacher trained toward convergence is very likely to overfit its soft predictions on the distillation data. Intuitively, it is possible to tackle this problem by early stopping the teacher training (Cho & Hariharan, 2019). However, a meticulous hyperparameter search may be required since the epoch number to find the best checkpoint is often sensitive to the specific training setting such as the learning rate schedule. It is also possible to save multiple early teacher checkpoints for the student to be distilled from sequentially (Jin et al., 2019). Additionally, one can utilize a "cross-fitting" procedure to prevent teacher from memorizing the training data. Namely, the training data is first partitioned into several folds, where the teacher predictions on each fold are generated by the teacher trained only on out-of-fold data (Dao et al., 2021). However, this may depart from the standard teacher training practice and induce significant computation overhead. One can also train the teacher network jointly with student's network blocks, which imposes a regularization toward the student performance (Park et al., 2021). However, this may also induce significant computation overhead as every unique student architecture will require retraining the teacher. For multi-generation knowledge distillation specifically (Furlanello et al., 2018), one can penalize the difference between the predictive probability at the true class and the probabilities at other semantically relevant classes to encourage the learning of secondary probabilities in teacher's prediction, which is shown to improve the performance of models in later generations (Yang et al., 2019). Different from these attempts, we train the teacher to directly learn the true label distribution of its training data, leading to a simple and practical student-oriented teacher training framework with minimum computation overhead.

**Uncertainty learning on unseen data.** Since the objective of our student-oriented teacher training is to learn label distributions of the training data, it is related to those methods aiming to learn quality uncertainty on the unseen data. We consider those methods that are feasible for large teacher network training, including (1) classic approaches to overcome overfitting such as $\ell_1$ and $\ell_2$ regularization, (2) modern regularizations such as label smoothing (Szegedy et al., 2016) and data augmentations such as mixup (Zhang et al., 2018a) and Augmix (Hendrycks et al., 2020), (3) Post-training methods such as temperature scaling (Guo et al., 2017), as well as (4) methods that incorporate uncertainty as a learning ojective such as confidence-aware learning (CRL) (Moon et al., 2020).

We have conducted experiments on CIFAR-100 using all these methods and the results can be found in Appendix F. Unfortunately, the performance of these regularization methods is unsatisfactory in knowledge distillation — only CRL can slightly outperform the standard training. We believe the reasons might be two-folds. First, most existing criteria for uncertainty quality on the unseen data such as calibration error (Naeini et al., 2015) or ranking error (Geifman et al., 2019), only require the model to output an uncertainty estimate that is correlated with the probability of prediction errors. Such criteria may not be translated into the approximation error to the true label distribution. Second, even if a model learns true label distribution on unseen data, it does not necessarily have to learn true label distribution on the training data, as deep neural networks tend to memorize the training data.

## 7 CONCLUSION AND FUTURE WORK

In this work, we rigorously studied the feasibility to learn the true label distribution of the training data under a standard empirical risk minimization framework. We also explore possible improvements of current teacher training that facilitate such learning. In the future, we plan to adapt our theory to other knowledge distillation scenarios such as transfer learning and mixture of experts, and explore more effective student-oriented teacher network training methods.

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

## A   PROOF OF THEOREM 3.6

**Step I: Bound the difference between the true label distribution and the sample mean.**   First we note that the set of all training inputs can be grouped into several subsets such that the inputs in each subset possess similar true label distribution.

More formally, Let $\mathcal{C}^r = \{\bar{x}_j\}_{j=1}^{N_r}$ be an $r$-external covering of $\mathcal{S}$ with minimum cardinality, namely

$$\mathcal{S} \subseteq \bigcup_{x \in \mathcal{C}^r} \{x' \mid \|x' - x\| \le r\}, \tag{7}$$

where we refer $\bar{x}_j$ as the covering center, and $N_r$ is the covering number.

Let $\{\mathcal{S}_j^r\}_{j=1}^{N_r}$ be any disjoint partition of $\mathcal{S}$ such that $\mathcal{S}_j^r \subseteq \{x' \mid \|x' - \bar{x}_j\| \le r\}$. We show that $\mathcal{S}_j^r$ attains a property that the true label distribution $\mathbf{p}^*(x)$ of any input $x$ in this subset will not be too far from the sample mean of one-hot labels $\bar{\mathbf{y}}_j = |\mathcal{S}_j^r|^{-1} \sum_{x \in \mathcal{S}_j^r} \mathbf{1}_y$ in this subset. Namely given any $\varepsilon > 0$, we have

$$P\left(\|\mathbf{p}^*(x) - \bar{\mathbf{y}}_j\| \ge \varepsilon\right) \le 2 \exp\left(-\frac{|\mathcal{S}_j^r|}{2K}\tilde{\varepsilon}^2\right), \tag{8}$$

where $\tilde{\varepsilon} = \max(\varepsilon - 2L^*r, 0)$.

To prove this property we first present two lemmas.

**Lemma A.1** (Lipschitz constraint of the true label distribution). *Let $\mathcal{S}_j^r$ be a subset constructed above and $\bar{\mathbf{y}}_j = |\mathcal{S}_j^r|^{-1} \sum_{x \in \mathcal{S}_j^r} \mathbf{1}_y$. Then for any $x \in \mathcal{S}_j^r$ we have,*

$$\|\mathbf{p}^*(x) - \mathbb{E}[\bar{\mathbf{y}}_j]\| \le 2L^*r. \tag{9}$$

*Proof.* First, since $x \in \mathcal{S}_j^r$, we have $\|x - \bar{x}_j\| \le r$, which implies $\|\mathbf{p}^*(x) - \mathbf{p}^*(\bar{x}_j)\| \le L^*r$ by the locally Lipschitz continuity of $\mathbf{p}^*$. Then for any $x, x' \in \mathcal{S}_j^r$, we will have $\|\mathbf{p}^*(x) - \mathbf{p}^*(x')\| \le 2L^*r$ by the triangle inequality. Let $N_\mathcal{S} = |\mathcal{S}_j^r|$. Therefore,

$$\left\|\mathbf{p}^*(x) - \frac{1}{N_\mathcal{S}}\sum_{x \in \mathcal{S}_j^r} \mathbf{p}^*(x)\right\| \le 2\frac{N_\mathcal{S} - 1}{N_\mathcal{S}}L^*r \le 2L^*r. \tag{10}$$

Further, the linearity of the expectation implies

$$\mathbb{E}[\bar{\mathbf{y}}] = N_\mathcal{S}^{-1}\sum_{x \in \mathcal{S}_j^r} \mathbb{E}[\mathbf{1}_{y(x)}] = N_\mathcal{S}^{-1}\sum_{x \in \mathcal{S}_j^r} \mathbf{p}^*(x). \tag{11}$$

Therefore $\|\mathbf{p}^*(x) - \mathbb{E}[\bar{\mathbf{y}}_j]\| \le 2L^*r$. $\qquad\square$

**Lemma A.2** (Concentration inequality of the sample mean). *Let $\mathcal{S}$ be a set of $x$ with cardinality $N$. Let $\bar{\mathbf{y}} = N^{-1}\sum_{x \in \mathcal{S}} \mathbf{1}_y$ be the sample mean. Then for any $p$-norm $\|\cdot\|$ and any $\varepsilon > 0$, we have*

$$P\left(\|\bar{\mathbf{y}} - \mathbb{E}[\bar{\mathbf{y}}]\| \ge \varepsilon\right) \le 2 \exp\left(-\frac{N}{2K}\varepsilon^2\right). \tag{12}$$

*Proof.* Note that $\bar{\mathbf{y}}$ obeys a multinomial distribution, *i.e.* $\bar{\mathbf{y}} \sim N^{-1}multinomial(N, \mathbb{E}[\bar{\mathbf{y}}])$. We utilize the classic results on the concentration properties of multinomial distribution based on $\ell_1$ norm (Weissman et al., 2003; Qian et al., 2020), namely

$$P(\|\bar{\mathbf{y}} - \mathbb{E}[\bar{\mathbf{y}}]\|_1 \ge \varepsilon) \le 2 \exp\left(-\frac{N}{2K}\varepsilon^2\right) \tag{13}$$

Further, for any $\mathbf{q} \in \mathbb{R}^K$, by the equivalence of $p$-norm we have $\|\mathbf{q}\| \le \|\mathbf{q}\|_1$. Therefore,

$$\{\mathbf{q} \mid \|\mathbf{q}\| \ge \varepsilon\} \subseteq \{\mathbf{q} \mid \|\mathbf{q}\|_1 \ge \varepsilon\}, \tag{14}$$

which implies,

$$P\left(\|\mathbf{q}\| \ge \varepsilon\right) \le P\left(\|\mathbf{q}\|_1 \ge \varepsilon\right). \tag{15}$$

Combine inequality (13) and (15) we then have (12). $\qquad\square$

One can see that Lemma A.1 bounds the difference between true label distribution of individual inputs and the mean true label distribution, while Lemma A.2 bounds the difference between the sample mean and the mean true label distribution. Therefore the difference between the true label distribution and the sample mean is also bounded, since by the triangle inequality we have

$$\|\mathbf{p}^*(x) - \bar{\mathbf{y}}\| \le \|\mathbf{p}^*(x) - \mathbb{E}[\bar{\mathbf{y}}]\| + \|\bar{\mathbf{y}} - \mathbb{E}[\bar{\mathbf{y}}]\|. \tag{16}$$

This implies

$$P\left(\|\mathbf{p}^*(x) - \bar{\mathbf{y}}\| \ge \varepsilon\right) \le P\left(\|\bar{\mathbf{y}} - \mathbb{E}[\bar{\mathbf{y}}]\| \ge \tilde{\varepsilon}\right), \tag{17}$$

where $\tilde{\varepsilon} = \max(\varepsilon - 2L^*r, 0)$. Apply Lemma A.2 we then have (8).

**Step II: Bound the difference between the prediction given by the empirical risk minimizer and the sample mean.** Here we show that given the locally Lipschitz constraint established in each disjoint partition we constructed above, the prediction given by the empirical risk minimizer will be close to the sample mean. As an example, we focus on the negative log-likelihood loss, namely $\ell(\mathbf{f}(x), y) = -\mathbf{1}_y \cdot \log \mathbf{f}(x)$. Other loss functions that are subject to the proper scoring rule can be investigated in a similar manner.

First, we regroup the sum in (1) based on the partition constructed above, namely

$$\hat{R}(\mathbf{f}, \mathcal{S}) = \frac{1}{N_r} \sum_{j=1}^{N_r} \hat{R}(\mathbf{f}, \mathcal{S}_j^r), \tag{18}$$

where $\hat{R}(\mathbf{f}, \mathcal{S}_j^r) = -|\mathcal{S}_j^r|^{-1} \sum_{i=1}^{|\mathcal{S}_j^r|} \mathbf{1}_{y_i} \cdot \log \mathbf{f}(x_i)$ is the empirical risk in each partition. Since we are only concerned with the existence of a desired minimizer of the empirical risk, we can view $\mathbf{f}$ as able to achieve any labeling of the training inputs that suffices the local Lipschitz constraint. Thus the empirical risk minimization is equivalent to the minimization of the empirical risk in each partition. The problem can thus be defined as, for each $j = 1, \cdots, N_r$,

$$\begin{aligned} &\min_{\mathbf{f}} \hat{R}(\mathbf{f}, \mathcal{S}_j^r) \\ &s.t. \ \|\mathbf{f}(x) - \mathbf{f}(\bar{x}_j)\| \le Lr, \ \forall \ x \in \mathcal{S}_j^r, \end{aligned} \tag{19}$$

where the constraint is imposed by the locally-Lipschitz continuity of $\mathbf{f}$. By the following lemma, we show that the minimizer of such problem is achieved only if $\mathbf{f}(\bar{x}_j)$ is close to the sample mean.

**Lemma A.3.** *Let* $\bar{\mathbf{y}} = |\mathcal{S}_j^r|^{-1} \sum_{x \in \mathcal{S}_j^r} \mathbf{1}_y$. *The minimum of the problem (19) is achieved only if* $\mathbf{f}(\bar{x}_j) = \bar{\mathbf{y}}_\mathbf{j}(1 + KLr) - Lr$.

*Proof.* We note that since the loss function we choose is strongly convex, to minimize the empirical risk, the prediction of any input $x$ must be as close to the one-hot labeling as possible. Therefore the problem (19) can be formulated into a vector minimization where we can employ Karush–Kuhn–Tucker (KKT) theorem to find the necessary conditions of the minimizer.

We rephrase the problem (19) as

$$\begin{aligned} &\min_{\{\mathbf{p}_i\}_{i=1}^N} -\frac{1}{N} \sum_i \mathbf{1}_{y_i} \cdot \log \mathbf{p}_i \\ &s.t. \ \|\mathbf{p}_i - \mathbf{p}\| \le \varepsilon, \ \sum_k \mathbf{p}_i^k = 1, \ \sum_k \mathbf{p}^k = 1, \mathbf{p}_i^k \ge 0, \mathbf{p}^k \ge 0. \end{aligned} \tag{20}$$

**Case I.** We first discuss the case when $\mathbf{p}^k + \varepsilon < 1$ for all $k$. First, we observe that for any $\mathbf{p}$, the minimum of the above problem is achieved only if $\mathbf{p}_i^{y_i} = \mathbf{p}^{y_i} + \varepsilon$. Because by contradiction, if $\mathbf{p}_i^{y_i} < \mathbf{p}^{y_i} + \varepsilon$, we will have $-\log \mathbf{p}_i^{y_i} > -\log(\mathbf{p}^{y_i} + \varepsilon)$, and $\mathbf{p}^{y_i} + \varepsilon$ belongs to the feasible set, which means $\mathbf{p}_i^{y_i}$ does not attain the minimum.

The above problem can then be rephrased as

$$\min_{\mathbf{p}} -\frac{1}{N} \sum_i \log(\mathbf{p}^{y_i} + \varepsilon), \ s.t. \ \sum_k \mathbf{p}^k = 1, \mathbf{p}^k \ge 0, \tag{21}$$

where we have neglected the condition associated with $\mathbf{p}_i^{k \neq y_i}$, since they do not contribute to the objective, they can be chosen arbitrarily as long as the constraints are sufficed, and clearly the constraints are underdetermined.

Let $N_k = \sum_i 1(y_i = k)$, we have $\sum_i \log(\mathbf{p}^{y_i} + \varepsilon) = \sum_k N_k \log(\mathbf{p}^k + \varepsilon)$. Therefore the above problem is equivalent to

$$\min_{\mathbf{p}} - \sum_k \bar{\mathbf{y}}^k \log(\mathbf{p}^k + \varepsilon), \ \ s.t. \ \sum_k \mathbf{p}^k = 1, \mathbf{p}^k \geq 0, \tag{22}$$

where $\bar{\mathbf{y}} \equiv [N_1/N, \cdots, N_k/N]^T$ is equal to the sample mean $N^{-1} \sum_i \mathbf{1}_{y_i}$.

To solve the strongly convex minimization problem (22) it is easy to employ KKT conditions to show that

$$\mathbf{p} = \bar{\mathbf{y}}(1 + K\varepsilon) - \varepsilon.$$

**Case II.** We now discuss the case when $\hat{\mathbf{p}}$ is the minimizer of (20) and there exists $k'$ such that $\hat{\mathbf{p}}^{k'} + \varepsilon \geq 1$. And $\hat{\mathbf{p}} \neq \mathbf{p}^*$, where $\mathbf{p}^* = \bar{\mathbf{y}}(1 + K\varepsilon) - \varepsilon$ is the form of the minimizer in the previous case.

Considering a non-trivial case $\mathbf{p}^{*k'} < 1 - \varepsilon$. Otherwise the true label distribution is already close to the one-hot labeling, which is the minimizer of the empirical risk. Therefore by $\sum_{k \neq k'} \mathbf{p}^{*k} > \varepsilon$ we have the condition

$$\sum_{k \neq k'} \bar{\mathbf{y}}^k > \frac{K\varepsilon}{1 + K\varepsilon} \tag{23}$$

Now considering the minimization objective $R(p) = -N^{-1} \sum_i \mathbf{1}_{y_i} \cdot \log \mathbf{p}_i$. For all $i$ with $y_i = k'$, we must have $\mathbf{p}_i^{y_i} = 1$, otherwise the optimal cannot be attained by contradiction. Then the minimization problem can be rephrased as

$$\min \sum_{k \neq k'} \bar{\mathbf{y}}^k \log(\hat{\mathbf{p}} + \varepsilon), \ \ s.t. \ \sum_{k' \neq k} \hat{\mathbf{p}}^k \geq \varepsilon, \hat{\mathbf{p}}^k \geq 0, \tag{24}$$

where the first constraint is imposed by $\hat{\mathbf{p}}^{k'} \geq 1 - \varepsilon$.

Employ KKT conditions similarly we can have $\hat{\mathbf{p}}^k = \bar{\mathbf{y}}^k/\lambda - \varepsilon$ where $\lambda$ is a constant. By checking the constraint we can derive $\lambda \geq \sum_k \bar{\mathbf{y}}^k/(K\varepsilon)$.

However, the minimization objective

$$\min_{\lambda} - \sum_{k \neq k'} \bar{\mathbf{y}}^k \log \frac{\bar{\mathbf{y}}^k}{\lambda},$$

requires $\lambda$ to be minimized. Therefore $\lambda = \sum_{k \neq k'} \bar{\mathbf{y}}^k/(K\varepsilon)$, which implies

$$\hat{\mathbf{p}}^k = K\varepsilon \frac{\bar{\mathbf{y}}^k}{\sum_{k \neq k'} \bar{\mathbf{y}}^k} - \varepsilon. \tag{25}$$

Now since $\hat{\mathbf{p}} = \arg \min_p R(p)$ and $\hat{\mathbf{p}} \neq \mathbf{p}^*$, we must have $R(\hat{\mathbf{p}}) < R(\mathbf{p}^*)$. This means

$$-\sum_{k \neq k'} \bar{\mathbf{y}}^k \log \frac{K\varepsilon \bar{\mathbf{y}}^k}{\sum_{k \neq k'} \bar{\mathbf{y}}^k} < -\sum_{k \neq k'} \bar{\mathbf{y}}^k \log[\bar{\mathbf{y}}^k(1 + K\varepsilon)], \tag{26}$$

which is reduced to

$$\sum_{k \neq k'} \bar{\mathbf{y}}^k < \frac{K\varepsilon}{1 + K\varepsilon} \tag{27}$$

But this is contradict to our assumption. $\qquad \square$

We are now be able to bound the difference between the predictions of the training inputs produced by the empirical risk minimizer and the sample mean in each $\mathcal{S}_j^r$. To see that we have for each $x \in \mathcal{S}_j^r$.

$$
\begin{aligned}
\|\mathbf{f}(x) - \bar{\mathbf{y}}_j\| &\leq \|\mathbf{f}(x) - \mathbf{f}(\bar{x}_j)\| + \|\mathbf{f}(\bar{x}_j) - \bar{\mathbf{y}}_j\| \\
&\leq Lr(1 + K\|\bar{\mathbf{y}}_j - K^{-1}\mathbf{1}\|) \\
&\leq Lr(1 + K\|\mathbf{1}_{(\cdot)} - K^{-1}\mathbf{1}\|).
\end{aligned}
\tag{28}
$$

By Equation (8) we then have for any $x \in \mathcal{S}_j^r$,

$$
P\left(\|\mathbf{f}(x) - \mathbf{p}^*(x)\| \geq \varepsilon\right) \leq 2\exp\left(-\frac{|\mathcal{S}_j^r|}{2K}\tilde{\varepsilon}^2\right),
\tag{29}
$$

which means the difference between the predictions and the true label distribution is also bounded. Here $\tilde{\varepsilon} = \max(\varepsilon - Lr(1 + K\|\mathbf{1}_{(\cdot)} - K^{-1}\mathbf{1}\|) - 2L^*r, 0)$.

**Step III: Show the disjoint partition is non-trivial.** In (29), we have managed to bound the difference between the predictions yielded by an empirical risk minimizer and the true label distribution based on the cardinality of the subset $|\mathcal{S}_j^r|$, namely the number of inputs in $j$-partition. However $|\mathcal{S}_j^r|$ is critical to the bound here as if $|\mathcal{S}_j^r| = 1$, then (29) becomes a trivial bound. Here we show $|\mathcal{S}_j^r|$ is non-negligible based on simple combinatorics.

**Lemma A.4.** *Let $\{\mathcal{S}_j^r\}_{j=1}^{N_r}$ be a disjoint partition of the entire training set $\mathcal{S}$. Denote $\mathcal{S}^r(x)$ as the partition that includes $x$. Let $N(x) = |\mathcal{S}^r(x)|$ and $N = |\mathcal{S}|$. Then for any $\kappa \geq 1$,*

$$
\left|\left\{x \mid N(x) \geq \frac{N}{\kappa N^r}\right\}\right| \geq \left(1 - \frac{1}{\kappa} + \frac{1}{\kappa N^r}\right)N.
\tag{30}
$$

*Proof.* We note that the problem is to show the minimum number of $x$ such that $N(x) \geq N/(\kappa N^r)$. This is equivalent to find the maximum number of $x$ such that $N(x) \leq N/(\kappa N^r)$. Since we only have $N^r$ subsets, the maximum can be attained only if for $N^r - 1$ subsets $\mathcal{S}^r$, $|\mathcal{S}^r| = N/(\kappa N^r)$. Otherwise, if for any one of these subsets $|\mathcal{S}^r| < N/(\kappa N^r)$, then it is always feasible to let $|\mathcal{S}^r| = N/(\kappa N^r)$ and the maximum increases. Similarly, if the number of such subsets is less than $N^r - 1$, then it is always feasible to let another subset subject to $|\mathcal{S}^r| = N/(\kappa N^r)$ and the maximum increases. We can then conclude that at most $N(N^r - 1)/(\kappa N^r)$ inputs can have the property $N(x) \leq N/(\kappa N^r)$. $\square$

The above lemma basically implies when partitioning $N$ inputs into $N^r$ subsets, a large fraction of the inputs will be assigned to a subset with cardinality at least $N/(\kappa N^r)$. Here $N^r$ is the covering number and is bounded above based on the property of the covering in the Euclidean space. Apply Lemma A.4 to (29) we then arrive at Theorem 3.6.

**Difference from the Lipschitz bound in generalization theory.** We note that in Theorem 3.6, the constraint on the local Lipschitz continuity of the hypothesis function is more strict than that is required in typical generalization theory of deep neural networks (Sokolic et al., 2017; Bartlett et al., 2017).

**Remark A.5.** *In generalization theory we often require the hypothesis function to be locally Lipschitz continuous around training inputs, such that the difference between the prediction of an unseen input and the prediction of its nearest training input is bounded. However, for learning true label distribution of the training data we need to bound the difference between the predictions of a subset of training inputs. Therefore, our constraint requires the Lipschitz continuity to be established at a larger scale in the input space.*

Indeed, we observe from later experiments that our stronger Lipschitz constraint will actually hurt the performance of the teacher on the test set, whereas it can still improve the student performance.

## B  LIMITATIONS

In this paper we focus on the theoretical feasibility of learning the true label distribution of training examples with empirical risk minimization. Therefore we only analyze the existence of such a

desired minimizer, but neglect the optimization process to achieve it. By explore the optimization towards true label distribution, potentially more dynamics can be found to inspire new regularization techniques.

Also, our proposed method for training a student-oriented teacher may not be able to advance the state-of-the-art significantly, as the regularization techniques based inspired by our analyses such as Lipschitz regularization and consistency regularization may more or less leveraged by existing training practice of deep neural networks, either implicitly or explicitly.

## C    IMPLEMENTATION DETAILS

**Lipschitz regularization.**    Following previous practice using Lipschitz regularization for generalization on unseen data (Yoshida & Miyato, 2017) or stabilizing generative model (Miyato et al., 2018), we regularize the Lipschitz constant of a network by constraining the Lipschitz constant of each trainable component. The regularization term is thus defined as $\ell_{\text{LR}} = \sum_f \text{Lip}(f)$, where $f$ denotes a trainable component in the network $\boldsymbol{f}$. The Lipschitz constant of a network component $\text{Lip}(f)$ induced by a norm $\|\cdot\|$ is the smallest value $L$ such that for any input features $h, h'$, $\|f(h) - f(h')\| \leq L\|h - h'\|$. Here we adopt the Lipschitz constant induced by 1-norm, since its calculation is accurate, simple and efficient. For calculating the Lipschitz constants of common trainable components in deep neural networks, we refer to (Gouk et al., 2021) for a comprehensive study.

**Consistency regularization.**    We design our consistency regularization term as $\ell_{\text{CR}} = \frac{1}{N} \sum_i \left\| \boldsymbol{f}(x_i) - \overline{\boldsymbol{f}(x_i)} \right\|_2^2$, where we follow previous work (Laine & Aila, 2017) and employ MSE to penalize the difference. Here $\overline{\boldsymbol{f}(x)}$ is the aggregated prediction of an input $x$, which we calculate as the simple average of previous predictions $\overline{\boldsymbol{f}(x)}_t = \frac{1}{t} \sum_{t'=0}^{t-1} \boldsymbol{f}(x)_{t'}$, where we omit the data augmentation operator for simplicity. At epoch 0 we simply skip the consistency regularization. Note that such a prediction average can be implemented in an online manner thus there is no need to store every previous prediction of an input.

## D    DETAILS OF EXPERIMENT SETTING

### D.1    HYPERPARAMETER SETTING FOR TEACHER NETWORK TRAINING

For all the experiments on CIFAR-100, we employ SGD as the optimizer and train for 240 epochs with a batch size of 64. The learning rate is initialized at 0.05 and decayed by a factor of 10 at the epochs 150, 180 and 210, with an exception for ShuffleNet where the learning rate is initialized at 0.01 following existing practice (Tian et al., 2020; Park et al., 2021). The weight decay and momentum are fixed as 0.0005 and 0.9 respectively. The training images are augmented with random cropping and random horizontal flipping with a probability of 0.5.

For Tiny-ImageNet experiments, we employ SGD as the optimizer and conduct the teacher training for 90 epochs with a batch size of 128. The learning rate starts at 0.1 and is decayed by a factor of 10 at epochs 30 and 60. The weight decay and momentum are fixed as 0.0005 and 0.9 respectively. The training images are augmented with random rotation with a maximum degree of 20, random cropping and random horizontal flipping with a probability of 0.5. For student training the only difference is that we train for additional 10 epochs, with one more learning rate decay at epoch 90, aligned with previous settings (Tian et al., 2020).

For consistency regularization in our teacher training method, we experiment with various weight schedules besides the linear schedule mentioned in the main paper. We list the formulas for these schedules in the following. Here $t$ denotes the epoch number, $T$ denotes the total number of epochs, and $\lambda_{CR}^{\max}$ denotes the maximum weight.

- Cosine schedule:

$$\lambda_{CR}(t) = \cos\left[\left(1 - \frac{t}{T}\right)\frac{\pi}{2}\right]\lambda_{CR}^{\max}$$

Table 5: $\beta$ for different feature distillation algorithms

|         | $\beta$ | | | $\beta$ | | | $\beta$ | |
|---------|----------|-----------|-----|----------|-----------|-----|----------|-----------|
|         | Standard | SoTeacher | | Standard | SoTeacher | | Standard | SoTeacher |
| FitNets | 100      | 50        | AT  | 1000     | 500       | SP  | 3000     | 1500      |
| CC      | 0.02     | 0.01      | VID | 1.0      | 0.5       | RKD | 1.0      | 0.5       |
| PKT     | 30000    | 15000     | AB  | 1.0      | 0.5       | FT  | 200      | 100       |
| NST     | 50       | 25        | CRD | 0.8      | 0.5       |     |          |           |

- Cyclic schedule:

$$\lambda_{CR}(t) = \sqrt{1 - \left(1 - \frac{t}{T}\right)^2} \lambda_{CR}^{\max}$$

- Piecewise schedule:

$$\lambda_{CR}(t) = \begin{cases} 0, & 0 < t \leq T/3, \\ \lambda_{CR}^{\max}/2, & T/3 < t \leq 2T/3, \\ \lambda_{CR}^{\max}, & 2T/3 < t \leq T. \end{cases}$$

### D.2 HYPERPARAMETER SETTING FOR KNOWLEDGE DISTILLATION ALGORITHMS

For knowledge distillation algorithms we refer to the setting in RepDistiller [2]. Specifically, for original KD, the loss function used for student training is defined as

$$\ell = \alpha \ell_{\text{Cross-Entropy}} + (1 - \alpha)\ell_{KD}.$$

We grid search the best hyper-parameters that achieve the optimal performance, namely the loss scaling ratio $\alpha$ is set as $0.5$ and the temperature is set as $4$ for both CIFAR-100 and Tiny-ImageNet. For all feature distillation methods combined with KD the loss function can be summarized as (Tian et al., 2020)

$$\ell = \gamma \ell_{\text{Cross-Entropy}} + \alpha \ell_{KD} + \beta \ell_{\text{Distill}},$$

where we grid search the optimal $\gamma$ and $\alpha$ to be $1.0$ and $1.0$ respectively. When using our teacher training method, all these hyperparameters are kept same except that for all feature distillation algorithms the scaling weights corresponding to the feature distillation losses $\beta$ are cut by half, as we wish to rely more on the original KD that is well supported by our theoretical understanding. Table 5 list $\beta$ used in our experiments for all feature distillation algorithms. For SSKD (Xu et al., 2020) the hyperparameters are set as $\lambda_1 = 1.0$, $\lambda_2 = 1.0$, $\lambda_3 = 2.7$, $\lambda_4 = 10.0$ for standard training and $\lambda_1 = 1.0$, $\lambda_2 = 1.0$, $\lambda_3 = 1.0$, $\lambda_4 = 10.0$ for our methods. For the curriculum distillation algorithm RCO we experiment based on one-stage EEI (equal epoch interval). We select $24$ anchor points (or equivalently every 10 epochs) from the teacher's saved checkpoints.

## E ADDITIONAL EXPERIMENT RESULTS

## F EXPERIMENTS WITH UNCERTAINTY REGULARIZATION METHODS ON UNSEEN DATA

**Experiment setup.** We conduct experiments on CIFAR-100 with teacher-student pair WRN40-2/WRN40-1. We employ the original KD as the distillation algorithm. The hyperparameter settings are the same as those mentioned in the main results (see Appendix D). For each regularization method we grid search the hyperparameter that yields the best student performance. The results are summarized in Table 7.

**Classic regularization.** We observe that with stronger $\ell_2$ or $\ell_1$ regularization the student performance will not deteriorate significantly as teacher converges. However, it also greatly reduces the performance of the teacher. Subsequently the performance of the student is not improved as shown in Table 7.

---

[2]https://github.com/HobbitLong/RepDistiller

Table 6: *SoTeacher* consistently outperforms Standard on CIFAR-100 with various KD algorithms.

|  | WRN40-2/WRN40-1 | | WRN40-2/WRN16-2 | | ResNet32x4/ShuffleV2 | |
|---|---|---|---|---|---|---|
|  | Standard | *SoTeacher* | Standard | *SoTeacher* | Standard | *SoTeacher* |
| FitNet | $74.06 \pm 0.20$ | $\mathbf{74.88} \pm 0.15$ | $75.42 \pm 0.38$ | $\mathbf{75.64} \pm 0.20$ | $76.56 \pm 0.15$ | $\mathbf{77.91} \pm 0.21$ |
| AT | $73.78 \pm 0.40$ | $\mathbf{75.12} \pm 0.17$ | $75.45 \pm 0.28$ | $\mathbf{75.88} \pm 0.09$ | $76.20 \pm 0.16$ | $\mathbf{77.93} \pm 0.15$ |
| SP | $73.54 \pm 0.20$ | $\mathbf{74.71} \pm 0.19$ | $74.67 \pm 0.37$ | $\mathbf{75.94} \pm 0.20$ | $75.94 \pm 0.16$ | $\mathbf{78.06} \pm 0.34$ |
| CC | $73.46 \pm 0.12$ | $\mathbf{74.76} \pm 0.16$ | $75.08 \pm 0.07$ | $\mathbf{75.67} \pm 0.39$ | $75.43 \pm 0.19$ | $\mathbf{77.68} \pm 0.28$ |
| VID | $73.88 \pm 0.30$ | $\mathbf{74.89} \pm 0.19$ | $75.11 \pm 0.07$ | $\mathbf{75.71} \pm 0.19$ | $75.95 \pm 0.11$ | $\mathbf{77.57} \pm 0.16$ |
| RKD | $73.41 \pm 0.47$ | $\mathbf{74.66} \pm 0.08$ | $75.16 \pm 0.21$ | $\mathbf{75.59} \pm 0.18$ | $75.28 \pm 0.11$ | $\mathbf{77.46} \pm 0.10$ |
| PKT | $74.14 \pm 0.43$ | $\mathbf{74.89} \pm 0.16$ | $75.45 \pm 0.09$ | $\mathbf{75.53} \pm 0.09$ | $75.72 \pm 0.18$ | $\mathbf{77.84} \pm 0.03$ |
| AB | $73.93 \pm 0.35$ | $\mathbf{74.86} \pm 0.10$ | $70.09 \pm 0.66$ | $\mathbf{70.38} \pm 0.87$ | $76.27 \pm 0.26$ | $\mathbf{78.05} \pm 0.21$ |
| FT | $73.80 \pm 0.15$ | $\mathbf{74.75} \pm 0.13$ | $75.19 \pm 0.15$ | $\mathbf{75.68} \pm 0.28$ | $76.42 \pm 0.17$ | $\mathbf{77.56} \pm 0.15$ |
| NST | $73.95 \pm 0.41$ | $\mathbf{74.74} \pm 0.14$ | $74.95 \pm 0.23$ | $\mathbf{75.68} \pm 0.16$ | $76.07 \pm 0.08$ | $\mathbf{77.71} \pm 0.10$ |
| CRD | $74.44 \pm 0.11$ | $\mathbf{75.06} \pm 0.37$ | $75.52 \pm 0.12$ | $\mathbf{75.95} \pm 0.02$ | $76.28 \pm 0.13$ | $\mathbf{78.09} \pm 0.13$ |
| SSKD | $75.82 \pm 0.22$ | $\mathbf{75.94} \pm 0.18$ | $76.31 \pm 0.07$ | $\mathbf{76.32} \pm 0.09$ | $78.49 \pm 0.10$ | $\mathbf{79.37} \pm 0.11$ |
| RCO | $74.50 \pm 0.32$ | $\mathbf{74.81} \pm 0.04$ | $75.24 \pm 0.34$ | $\mathbf{75.50} \pm 0.12$ | $76.75 \pm 0.13$ | $\mathbf{77.59} \pm 0.31$ |

Table 7: Performance of the knowledge distillation when training the teacher using existing regularization methods for learning quality uncertainty on unseen data.

|  | WRN40-2/WRN40-1 | |
|---|---|---|
|  | **Student** | Teacher |
| Standard | $73.73 \pm 0.13$ | $76.38 \pm 0.13$ |
| *SoTeacher* | $\mathbf{74.35} \pm 0.23$ | $74.95 \pm 0.28$ |
| $\ell_2$ ($5 \times 10^{-4}$) | $73.73 \pm 0.13$ | $76.38 \pm 0.13$ |
| $\ell_1$ ($10^{-5}$) | $73.60 \pm 0.15$ | $73.52 \pm 0.05$ |
| Mixup ($\alpha = 0.2$) | $73.19 \pm 0.21$ | $77.30 \pm 0.20$ |
| Cutmix ($\alpha = 0.2$) | $73.61 \pm 0.26$ | $78.42 \pm 0.07$ |
| Augmix ($\alpha = 1, k = 3$) | $73.83 \pm 0.09$ | $77.80 \pm 0.30$ |
| CRL ($\lambda = 1$) | $74.13 \pm 0.29$ | $76.69 \pm 0.16$ |

**Label smoothing.**    Label smoothing is shown to not only improve the performance but also the uncertainty estimates of deep neural networks (Müller et al., 2019). However, existing works have already shown that label smoothing can hurt the effectiveness of knowledge distillation (Müller et al., 2019), thus we neglect the results here. An intuitive explanation is that label smoothing encourages the representations of samples to lie in equally separated clusters, thus "erasing" the information encoding possible secondary classes in a sample (Müller et al., 2019).

**Data augmentation.**    Previous works have demonstrated that mixup-like data augmentation techniques can greatly improve the uncertainty estimation on unseen data (Thulasidasan et al., 2019; Hendrycks et al., 2020). For example, Mixup augmented the training samples as $x := \alpha x + (1 - \alpha) x'$, and $y := \alpha y + (1 - \alpha) y'$, where $(x', y')$ is a randomly drawn pair not necessarily belonging to the same class as $x$.

As shown in Table 7, stronger mixup can improve the performance of the teacher, whereas it can barely improve or even hurt the performance of the student. Based on our theoretical understanding (Theorem 3.8), we conjecture the reason might be that mixup distorts the true label distribution of an input stochastically throughout the training, thus hampering the learning of true label distribution.

**Temperature scaling.**    Previous works have suggested using the uncertainty on a validation set to tune the temperature for knowledge distillation either in standard learning (Menon et al., 2021) or robust learning (Dong et al., 2021). However, the optimal temperature may not be well aligned with that selected based on uncertainty (Menon et al., 2021). We neglect the experiment results here as the distillation temperature in our experiments is already fine-tuned.

**Uncertainty learning.**    CRL designs the loss function as $\ell = \ell_{\text{CE}} + \lambda \ell_{\text{CRL}}$, where $\ell_{\text{CE}}$ is the cross-entropy loss and $\ell_{\text{CRL}}$ is an additional regularization term bearing the form of

$$\ell_{\text{CRL}} = \max \left( 0, -g(c(x_i), c(x_j))(p(x_i) - p(x_j)) + |c(x_i) - c(x_j)| \right), \quad (31)$$

where $p(x) = \max_k \mathbf{f}(x)^k$ is the maximum probability of model's prediction on a training sample $x$ and

$$c(x) = \frac{1}{t-1} \sum_{t'=1}^{t-1} 1(\arg \max_k \mathbf{f}(x)_{t'}^k = y)$$

is the frequency of correct predictions through the training up to the current epoch. Here $g(c_i, c_j) = 1$ if $c_i > c_j$ and $g(c_i, c_j) = -1$ otherwise. Although originally proposed to improve the uncertainty quality of deep neural networks in terms of ranking, we found that CRL with a proper hyperparameter can improve the distillation performance, as shown in Table 7.

We note that the effectiveness of CRL on distillation can be interpreted by our theoretical understanding, as its regularization term (31) is essentially a special form of consistency regularization. To see this we first notice (31) is a margin loss penalizing the difference between $p(x)$ and $c(x)$ in terms of ranking. We then rewrite $c(x)$ as

$$c(x) = \mathbf{1}_y \cdot \frac{1}{t-1} \sum_{t'=1}^{t-1} \text{Onehot}[\mathbf{f}(x)_{t'}], \tag{32}$$

which is similar to our consistency regularization target, except the prediction is first converted into one-hot encoding. Such variation may not be the best for knowledge distillation as we wish those secondary class probabilities in the prediction be aligned as well.

