# OpenReview forum: "SoTeacher: Toward Student-oriented Teacher Network Training for Knowledge Distillation"
_ICLR.cc/2023/Conference — Submitted to ICLR 2023_

### Official Review · Reviewer_3FzY · 2022-10-23

**Confidence:** 4
**Clarity, Quality, Novelty And Reproducibility:** See above.
**Correctness:** 2
**Technical Novelty And Significance:** 3
**Empirical Novelty And Significance:** 3
**Recommendation:** 5

**Strength And Weaknesses:**

#### **Strengths**

This submission tackles an important and well-motivated problem. They do extensive experiments to show the benefit of the proposed method. While I have some concerns about the validation of the results (see below), I find the experimental section mostly convincing. Most details for reproducing the work are presented. The related work is adequately cited. To my best knowledge the proposed approach is novel in the context of knowledge distillation.

#### **Weakness 1: the theoretical part of the paper is not rigorous enough**
In particular, some parts seem to be wrong; some non-trivial proofs are missing; some parts of existing proofs are hard to follow; and there are extra assumptions being made in the proofs that are not listed in the statements.

1. I understand that Sec. 3.1 serves as a warmup, but the definition 3.1 of siblings is not realistic. For real-world datasets, even with full knowledge of $p^*(x)$ the set of siblings for most examples will probably consist of only that example.
2. The proof of proposition 3.3 should not be omitted. I am not sure that the arguments presented for the negative log-likelihood loss hold for other proper scoring rules. Why should the optimal value of $f^*$ be achieved at the empirical average of one-hot labels for other loss functions? For example, if we take $\ell(p, q) = \max_{i \in [K]} |p_i - q_i|$ and consider a siblings set containing 3 examples with one-hot labels $\\{[0,1], [1,0], [1,0]\\}$, then it is easy to verify that the optimal value of $f^*$ will be $[1, 0]$, not $[2/3, 1/3]$.
3. On the main theorem (Thm. 3.6)
    a) Is it realistic to expect that $N^r$ is not of the scale of $N$ for real-world datasets? It seems that most examples in such datasets would be far from each other in the input space (measured by a p-norm).
    b) The theorem statement is for general loss functions that are proper scoring rules, but the proof is done for the negative log-likelihood. The authors mention that other loss functions can be investigated in a similar manner, but I am not sure about it. The Lemma A.3 relies on the form of the loss function heavily. Furthermore, as in the case of Proposition 3.3 the empirical loss minimizer in a neighborhood doesn’t need to be close to the empirical average of one-hot labels.
    c) Eq. (18) is not correct – an average of averages is not equal to the global average. It will be true if the partition subsets are of equal size.
    d) “Since we are only concerned with the existence of a desired minimizer of the empirical risk, we can view $f$ as able to achieve any labeling of the training inputs that suffices the local Lipschitz constraint”. Here the authors assume that the function class $\mathcal{F}$ is so large that empirical losses on different neighborhoods can be optimized separately. For this to happen one probably needs extra assumptions on $\mathcal{F}$. For example, this is not true for the set of linear functions.
    e) The proof of Lemma A.3 is hard to follow. I was not able to verify it. For example in the case I, should not *all* dimensions of $p^k$ be specified in order to verify whether $||p - p^k|| \le \epsilon$ holds or not. In general, do authors claim that the minimum of problem (19) is achieved at the same point no matter what p-norm is used in the constraints? I don’t see how this happens.

4. The proof of Thm 3.8 is omitted. It seems that the proof will be different as $f^*$ is minimizing the empirical risk on augmented examples. It is also not clear whether $N^r$ refers to the covering number with augmented neighborhoods or not.

#### **Weakness 2: validity of the results**
Training a teacher with early stopping and tuning the knowledge distillation temperature are two standard approaches that help alleviate overfitting the one-hot training labels. This paper does not compare the proposed method with early stopped teachers, justifying that the latter needs tedious hyperparameter tuning. I suggest comparing with teachers trained for 50%, 60%, …, or 100% of epochs.

The temperature parameter in knowledge distillation usually has a strong effect on the student's performance. It would be great to see if proper tuning of the temperature parameter can bring the same benefit as the proposed method.

#### **Minor comments**
* “And $\lVert \cdot \rVert$ is a distance function induced by an arbitrary p-norm”. This makes the paper unclear. I suggest picking a concrete p-norm early in the paper.
* Lipschitz regularization is similar to bounding gradient with respect to inputs x. This kind of regularization was explored in the context of training neural networks that are robust to small input changes (see for example [1]).
* Def 3.2: in equation (1) $\ell$ is defined on $(f(x), y)$ pairs, while in most of the text $\ell$ is defined on a pair of probability distributions over the classes.
* Notations like $x \in \mathcal{S}_{p^*}(x)$ should be avoided.
* Definition 3.5: change to “We say *a set of inputs* $\mathcal{C}$ is a … “.
* $N^r$ can be confused with taking the r-th power of N. You can use $N_r$ instead.
* In Def. 3.7  $\mathcal{A}$ is a set rather than a function.
* In Thm. 3.8, the maximums over a possibly not compact sets should be replaced with supremums.
* $[N] = \\{1,2,\ldots,N\\}$, rather than $[N] \equiv 1,2,...,N$.
* In the proof of Thm 4.6, $\mathcal{S}$ is not defined. I assume it refers to the set of training inputs.


#### **References**

[1] Hoffman J, Roberts DA, Yaida S. Robust learning with Jacobian regularization. arXiv preprint arXiv:1908.02729.


**Summary Of The Paper:**

This submission focuses on training teacher models with the goal of making them successful in knowledge distillation. The first part of the paper presents necessary conditions under which a teacher model minimizing the empirical loss computed with one-hot labels captures the Bayes classifier probabilities on the training set, thus providing better supervision in knowledge distillation. These theoretical observations indicate that it is desirable to have the teacher network be locally Lipschitz around the training examples and have consistent outputs under data augmentations of a single example. Based on these findings the authors propose a method of training a teacher model that includes Lipschitz regularization and consistency regularization. They show that this way of training produces teacher models that distill better students in various knowledge distillation procedures. The proposed method is also compared with some standard ways of teacher regularization schemes designed to prevent overfitting.

**Summary Of The Review:**

In summary, the quality and clarity of the theoretical parts of the paper need to be improved significantly. This is the main determinant of my score.

---

> ### Author Response · Authors · 2022-11-23
> **Replies to Reviewer 3FzY**
>
> Thanks for your valuable comments. Please check our replies below.
>
> **____**
>
> **Q. Definition of siblings in the real world**
>
> We agree with the reviewer that the definition of siblings in Section 3.1 (a hypothetical case) is not that realistic. This hypothetical case aims to demonstrate our core idea of learning true label distribution on empirical data. Based on this we can smoothly shift to the realistic case, where the definition of siblings can be simply relaxed, such that the sibling set can now contain examples with similar true label distributions instead of exactly same true label distribution.
>
> We also wish to add that, for realistic datasets, siblings defined in a strict sense may not exist is because it is hard to sample them, rather than because they don't exist. Consider, for simplicity, a 2D isotropic Gaussian mixture model with 2 centers where each class possesses one center. Then any examples that are equidistant to the 2 centers would be siblings. But it would be rare to sample them since the equidistant line has a zero measure.
>
>
> **____**
>
> **Q. Proof of Proposition 3.3**
>
> We note that Proposition 3.3 is a direct consequence of the property of proper scoring rules. Proper scoring rules are a specific type of loss function $l$ which will be minimized only if the predicative distribution is equal to the target distribution. Such a property ensures that the predicative distribution needs to be equal to the empirical expectation of all one-hot labels of siblings to minimize the empirical loss. Aside from the NLL loss mentioned in the paper, here we provide another example, namely the commonly used MSE loss. Following the notation in the paper, we can see that $\mathbb{\hat{E}} \|\mathbf{1}_y - \mathbf{f}(x)\|_2^2 = \|1 - \mathbb{\hat{E}}[\mathbf{1}_y]\|_2^2 + \|\mathbb{\hat{E}}[\mathbf{1}_y] - \mathbf{f}(x)\|_2^2 \ge \|1 - \mathbb{\hat{E}}[\mathbf{1}_y]\|_2^2$, with equality if and only if $\mathbf{f}(x) = \mathbb{\hat{E}}[\mathbf{1}_y]$. The first equation here is known as a decomposition of the proper scoring rule. For the decomposition of general proper scoring rules, we recommend [1] as a reference.
>
> We add that the loss example mentioned by the reviewer (max absolute error) does not comply with this property because it is not a proper scoring rule.

---

> > ### Author Response · Authors · 2022-11-23
> > **Replies to Reviewer 3FzY (Continued)**
> >
> > **____**
> >
> > **Q. Proof of Theorem 3.8**:
> >
> > This Theorem is a simple extension of Theorem 3.6 with data augmentations. We can still perform the covering for the original training examples, where augmented examples can be viewed as additional examples associated with each training example. Therefore, for Step II in the proof, we only have to first ensure the predictions of augmented examples are close to the original examples within a distance of $\varepsilon'$, which means they are close to the center prediction with a distance of $\varepsilon + \varepsilon'$. Then the problem reduces to the original problem albeit now $\varepsilon:= \varepsilon + \varepsilon'$. Step I would be the same as the neighborhood definition is unchanged, since we assume data augmentation will not significantly change the true label distribution. We will provide proof for this Theorem in the revision.
> >
> >
> > **____**
> >
> > **Q. Multiple intermediate teachers checkpoints for comparison.**
> >
> > Figure 1 in our paper has shown the student performances distilled from multiple intermediate checkpoints. There do exist early stopped teacher checkpoints that perform well for knowledge distillation, but it would be less feasible to identify it in practice. This is because first the teacher's own performance is no longer a reliable signal, and second if we use student performance as a signal, the computation cost would be huge as for every teacher checkpoint we need to train a student.
> >
> > **____**
> >
> > **Q. Proper tuning of the temperature.**
> >
> > We note that the temperature is indeed important to student's performance both in practice and in theory [2]. In the results reported in our paper, as also mentioned in Appendix F, the temperature is already fine-tuned for the baseline method, and is kept the same for our teacher regularization method. We wish to add that temperature scaling as a post-training processing can only modify the soft predictions of all training examples in whole but not their ranking, which thus may not perform as good as regularization during training.
> >
> >
> > [1] Reliability, Sufficiency, and the Decomposition of Proper Scores. Brocker et al., 2009.
> >
> > [2] A statistical perspective on distillation. Menon et al. 2021.

---

> > > ### Comment · Reviewer_3FzY · 2022-12-04
> > > **Response to the authors**
> > >
> > > Thank you for the detailed response.
> > >
> > > > Q. Definition of siblings in the real world
> > >
> > > Thanks for the clarification. I think we are not the same page here. I was also hinting that most of the time a sibling set will be a zero-measure set.
> > >
> > >
> > > > Q. Proof of Proposition 3.3
> > >
> > > Thanks for pointing out that the example I brought is invalid, since the loss function is not a proper scoring rule. I recommend to add a short proof of Proposition 3.3 anyway. This will help to highlight further the proper scoring rule assumption.
> > >
> > >
> > > > Q. Main theorem (Theorem 3.6), (a) Scale of $N^r$ compared to $N$
> > >
> > > I am not of the opinion that a covering with respect to a $p$-norm in the space of natural images can be small, while satisfying the property that in each cover set examples have roughly similar label distributions $p^*$.
> > >
> > >
> > > > Q. Main theorem (Theorem 3.6), (b) Other loss functions (Lemma A.3)
> > > > Q. Main theorem (Theorem 3.6), (c) Average of average (Eq. (18)
> > > > Q. Proof of Theorem 3.8
> > > > Q. Main theorem (Theorem 3.6), Proof of Lemma A.3
> > >
> > > Thanks for the clarifications. Since no revision was uploaded, I cannot verify these statements confidently.
> > >
> > > > Q. Main theorem (Theorem 3.6), (d) Separate optimization on neighborhoods
> > >
> > > I think that this is a strong assumption. I recommend the authors to discuss this assumption in future revisions in detail, possibly providing evidence for this.
> > >
> > >
> > > > Q. Multiple intermediate teachers checkpoints for comparison.
> > >
> > > There is no need to compare with many teacher teachpoints. It will be enough to compare with 4-6 reasonable chosen checkpoints. The best student can be then selected using a small validation set.
> > >
> > > I understand that this procedure will be a few times computationally more expensive that the proposed method. Nevertheless, it is important to separate out the computational efficiency aspect, and check whether the proposed method outperforms early stopping.
> > >
> > >
> > > > Q. Proper tuning of the temperature.
> > >
> > > Thanks for the clarification.

---

> > ### Author Response · Authors · 2022-11-23
> > **Replies to Reviewer 3FzY (Continued)**
> >
> > **____**
> >
> > **Q. Main theorem (Theorem 3.6)**
> >
> > **(a) Scale of $N^r$ compared to $N$**: We believe it is realistic to expect training examples nearby to each other in the input space will have similar true label distribution. This means that an appropriate radius $r$ should exist to identify a neighborhood subset (defined in Section 3.2) where training examples share similar true label distribution, irrespective of the absolute distance between training examples. Therefore a cover with a radius $r$ should have a small covering number.
> >
> >
> > **(b) Other loss functions (Lemma A.3)**: We agree with the reviewer that the proof of Lemma A.3 relies on the specific form of NLL loss. It should be easy to see that MSE loss also complies with this lemma as it can be decomposed in a similar way such that the predicative distribution needs to be close to the mean of one-hot labels (see previous answers). We will modify the proof to accommodate general proper scoring rules in the revision. We appreciate the reviewer bringing this to our attention.
> >
> >
> > **(c) Average of average (Eq. (18))**: We agree with the reviewer that the equivalence between losses is not appropriate in Eq. (18). We intended to show their minimizer is equivalent, which is correct if the network function can achieve any labeling thus the minimization can be conducted in each partition. We thank the reviewer for pointing this out and we will modify it in the revision.
> >
> > **(d) Separate optimization on neighborhoods**: We believe for deep neural networks it is natural to assume the model can achieve any labeling of a fixed set of training examples. We will formulate this into an explicit assumption in the revision.
> >
> > **(e) Proof of Lemma A.3**: In the proof of this Lemma, because only the dimension corresponding to the label appears in the optimization objective, $\mathbf{p}_i[y]$ (here $i$ means the $i$-th training example, and $[\cdot]$ means the $\cdot$-th dimension of $\mathbf{p}$) needs to be as close to $1$ as possible to optimize it. But as the reviewer points out, to optimize the objective, here it is indeed not always possible that  $\mathbf{p}_i[y] = \mathbf{p}[y] + \varepsilon$ given the constraint $\|\mathbf{p}_i - \mathbf{p}\|\le \varepsilon$. We carefully reviewed this proof and found that it is dependent on the given norm. For $1$-norm, the best possible case would be $\mathbf{p}[y] = \mathbf{p}[y] + \varepsilon / 2$ while for $\infty$-norm, $\mathbf{p}[y] = \mathbf{p}[y] + \varepsilon$ is achievable. An arbitrary $p$-norm would be between these two cases. For example, for $2$-norm, $\mathbf{p}[y] = \mathbf{p}[y] + \varepsilon (K-1)/ K$, where $K$ is the number of dimensions of $\mathbf{p}$. So a correct statement should be $\mathbf{p}_i[y] = \mathbf{p}[y] + \varepsilon/\tau$, where $1\le \tau \le 2$. But one may see that this will only add a norm-dependent constant scale on the result of Lemma A.3 and won't affect the overall proof. We again thank the reviewer for bringing this to our attention.

---

### Official Review · Reviewer_FGnr · 2022-10-25

**Confidence:** 4
**Correctness:** 4
**Technical Novelty And Significance:** 3
**Empirical Novelty And Significance:** 3
**Recommendation:** 6

**Clarity, Quality, Novelty And Reproducibility:**

Clarity: Good. The paper is easy to follow and in good shape.

Quality: Good. The method, proof, and experiments are well-organized and clearly shown.

Novelty: Good. The research topic is novel.

Reproducibility: Fair. No supplementary code.

**Strength And Weaknesses:**

**Strengths**
+ The paper is well-written and easy to follow.
+ The motivation is clearly explained by theoretical analysis and examples.
+ The scope of the paper is novel and interesting that improving the training of teachers in the context of knowledge distillation can further boost the performance of multiple KD approaches.
+ The method are well demonstrated from both theoretical and empirical perspectives.
+ Comprehensive experiments and ablation studies with promising results well support the main claims and contributions of the paper.


**Weaknesses**

However, there are still some concerns to be addressed:

- That would be interesting to explore the influence of the proposed regularizers on reducing the overfitting of different teachers. How do they affect different teachers for the same student with KD methods?

- Table 6 looks to belong to the ablation study, however, it is shown in the appendix as "As shown in Table 6" in the main text. It is a bit confusing for readers.

- The lack of experiments on the large-scale dataset ImageNet and current transformer architectures weaken the paper.

**Summary Of The Paper:**

The paper is motivated by recent findings in the field of knowledge distillation that the teacher should learn the true label distribution on the inputs rather than the one-hot labels with the best performance. The authors propose to take advantage of the Lipschitz regularization and consistency regularization to train teachers that are better for distillation. The method is evidenced both theoretically and empirically in the image classification setting.

**Summary Of The Review:**

The paper is in good shape with both theoretical and empirical evidence for their proposed method. Moreover, the results of experiments on CIFAR-100 and TinyImageNet are sound to support the main contributions of the paper. However, it is a bit weak from the experiment side due to the lack of experiments on the large-scale ImageNet.

---

> ### Author Response · Authors · 2022-11-23
> **Replies to Reviewer FGnr**
>
> Thanks for your valuable comments. Please check our replies below.
>
>
> **_____**
>
> **Q. Effect of regularization on different teachers**
>
> We assume by "different teachers" the reviewer means "different teacher architectures". We conduct additional experiments on distilling from different teacher architectures (WRN-40-2 / ResNet32x4) into the same student architecture (ShuffleNet). As shown in the table, our teacher regularization methods can improve the distillation performance with different teacher architectures. The improvement is more significant for higher teacher performance, in which case the teacher may be more likely to overfit the one-hot labels.
>
> ||ResNet32x4 -> ShuffleNet||WRN40-2 -> ShuffleNet||
> |-|-|-|-|-|
> ||**Student Acc (%)**|**Teacher Acc (%)**|**Student Acc (%)**|**Teacher Acc (%)**|
> | Standard | $74.86\pm 0.18$ | $79.22\pm 0.03$ |  $76.05\pm 0.04$ | $76.20\pm 0.51$|
> | SoTeacher |  $77.24\pm 0.09$ | $78.49\pm 0.09$ |  $76.99\pm 0.22$ | $74.95\pm 0.28$$|
> | LR-only |  $76.52\pm 0.52$ | $77.73\pm 0.17$ | $76.65\pm 0.34$ | $74.30\pm 0.12$|
> | CR-only |  $76.23\pm 0.18$ | $80.01\pm 0.18$  | $76.75\pm 0.30$ | $76.71\pm 0.16$|

---

### Official Review · Reviewer_fjbL · 2022-10-25

**Confidence:** 3
**Correctness:** 3
**Technical Novelty And Significance:** 3
**Empirical Novelty And Significance:** 3
**Recommendation:** 6

**Clarity, Quality, Novelty And Reproducibility:**

Good novelty. Encouraging empirical results. Well written but need to be more details to be reproducible.

**Strength And Weaknesses:**

Strengths:

1. New study on feasibility of true label distribution.

2. Theoretical analysis is supported with empirical results when proposed regularisation and consistency losses proposed improve the student performance which is consistent with theory.

3. Paper is well-written.

Weaknesses:

1. It would be good to detail the regularization and consistency losses $l_{LR}$ and $l_{CR}$.

2. Is Fig. 1 with total loss (including  $l_{LR}$ and $l_{CR}$) or only the original loss? I am also interested in how the Lipschitz regularisation loss and consistency loss change over-time when training SoTeacher? Beside the theory, it is also interesting to understand empirically how these proposed losses influence on the learning of the teacher (e.g., reduce overfitting)?

3. Looking at Fig. 1, it is interesting that if the standard teacher is well early stop (e.g., epoch 150), the student performance can be also very comparable, and the training of both teacher and distillation be reduced a lot?

**Summary Of The Paper:**

This work studies the knowledge distillation in the context of multi-class classification, and this paper aims to train a good teacher for it by exploring the feasibility of training a teacher that is oriented toward student performance with an empirical risk minimisation framework. Inspired from recent findings that the good teacher is one whose capability to approximate the true label distribution. The paper investigates the feasibility by beginning with siblings theory, saying in a hypothetical case where the siblings of an input are known, the empirical minimiser can produce predictions close to the true label distribution on the training data. With this siblings,, the paper proves that if the loss function is properly selected (e.g., with proper scoring rules) and the network is properly regularised, the close approximation of the true label distribution can be achieved. To be practical, the work shows the siblings can be relaxed in terms of neighbourhoods with the assumption of L locally-Lipschitz continuous of true labels and  the constraints of the network has to be locally-Lipschitz continuous. When the augmentation is employed, with assumption that the true label distribution is nearly invariant under data augmentations, this neighbourhood is relaxed for augmented inputs with augmented neighbourhood. Based on theoretical analysis, the paper proposes the training teacher method incorporating Lipschitz regularisation and consistency regularisation to improve the empirical risk minimisation.

**Summary Of The Review:**

Overall, the paper is well-written and a nice theory with empirical support. The results are encouraging as being tested on some benchmark datasets and with a number of network baselines. So the proposed method mostly outperforms the standard distillation. Ablation study is also provided.

---

> ### Author Response · Authors · 2022-11-23
> **Replies to Reviewer fjbL**
>
> Thanks for your valuable comments. Please check our replies below.
>
> **____**
>
> **Q. Details of the regularization and consistency losses**
>
> We have included the implementation details of the regularization and consistency losses in Appendix C. We will also release our implementation.
>
>
> **____**
>
> **Q. Figure 1 for LR and CR only**
>
> In Figure 1 we report the results with the combined loss. Here we also report the teacher and student performances at intermediate checkpoints for training with LR and CR losses only. We can see that both LR and CR can alleviate teacher overfitting, among which LR can alleviate it more significantly.
>
> ||Standard||SoTeacher||LR-only||CR-only||
> |-|- |- |- |- |- |- |- |- |
> |**Epoch**|**Student Acc (%)**| **Teacher Acc (%)**|**Student Acc (%)**| **Teacher Acc (%)**|**Student Acc (%)**| **Teacher Acc (%)**|**Student Acc (%)**| **Teacher Acc (%)**|
> | 150 | $74.45\pm 0.22$ | $74.05\pm 0.47$ | $73.67\pm 0.11$ | $71.24\pm 0.31$ | $73.66\pm 0.12$ | $71.69\pm 0.22$ | $74.26\pm 0.27$ | $74.10\pm 0.30$ |
> | 170 | $73.71\pm 0.24$ | $73.49\pm 0.26$ | $73.77\pm 0.14$ | $71.90\pm 0.18$ | $73.18\pm 0.13$ | $71.03\pm 0.44$ | $74.12\pm 0.15$ | $74.21\pm 0.19$ |
> | 190 | $74.35\pm 0.08$ | $76.36\pm 0.12$ | $74.52\pm 0.14$ | $74.75\pm 0.54$ | $74.49\pm 0.26$ | $74.50\pm 0.21$ | $74.25\pm 0.26$ | $76.47\pm 0.20$ |
> | 210 | $73.70\pm 0.05$ | $76.39\pm 0.16$ | $74.63\pm 0.08$ | $75.06\pm 0.39$ | $74.45\pm 0.17$ | $74.14\pm 0.17$ | $74.30\pm 0.21$ | $76.58\pm 0.28$ |
> | 230 | $73.82\pm 0.24$ | $76.40\pm 0.08$ | $74.73\pm 0.07$ | $74.99\pm 0.32$ | $74.44\pm 0.12$ | $74.40\pm 0.15$ | $74.08\pm 0.12$ | $76.71\pm 0.16$ |
> | 240 | $73.73\pm 0.13$ | $76.38\pm 0.13$ | $74.35\pm 0.23$ | $74.95\pm 0.28$ | $74.34\pm 0.11$ | $74.30\pm 0.12$ | $73.81\pm 0.15$ | $76.71\pm 0.16$ |
>
>
> **____**
>
> **Q. Compared with early stop teacher**
>
> We note that in Figure 1, there do exist early stopped teacher checkpoints that perform well for knowledge distillation, but it would be less feasible to identify it in practice. This is because first the teacher's own performance is no longer a reliable signal of the student performance, and second if we use student performance as a signal, the computation cost would be huge as for every teacher checkpoint during training we need to train a student.

---

### Official Review · Reviewer_gTte · 2022-10-29

**Confidence:** 4
**Correctness:** 1
**Technical Novelty And Significance:** 1
**Empirical Novelty And Significance:** 2
**Recommendation:** 3

**Clarity, Quality, Novelty And Reproducibility:**

As written in the weaknesses, I find this paper to be of limited novelty. I think the clearest novel contribution of the paper is the idea that Lipschitz regularisation may yield teachers that improve student performance, but this alone is insufficient for consideration in my opinion.

Clarity: I think that the mathematical statements are not written carefully enough, which leads to meaningful confusion.

For instance, in Section 3, the statements say "Let $\mathcal{F}$ be a family of functions that are $L$-locally-Lipschitz...", but this doesn't work - for instance, $\mathcal{F} = \\{ \mathbf{1}(x)/K\}$ (i.e. the single constant function that spits out the uniform law) is such a family (for $p^*$ nonconstant). I suppose that the intended statement is "Let $\mathcal{F}$ be **the** family of **all** $L$-locally-Lipschitz..." This is an important distinction! Otherwise one doesn't even know that there exists any $f \in \mathcal{F}$ such that $f(x) \approx p^*(x)$.

Similarly, local-Lipschitzness is never defined, symbols are used inconsistenly ($x \in \mathcal{S}_{p^*}(x)$) in Proposition 3.3. I think that the paper would benefit from a clearer writing of the mathematical parts.

The remainder of the paper is fairly clearly written, modulo the fact that a clear connection between the hypothesis of improved estimation of $p^*$ via Lipschitz regularisation is not really made.


Minor comments:

- Note also that the statements in Section 3.3 as stated are not uniform - the bound is true for every ball in the cover, but not simultaneously true for every ball (this is the usual distinction between pointwise and uniform laws of large numbers). You would need a $\log N^r$ in the error bound to extend the statement to all $x \in \tilde{\mathcal{D}}_{\mathcal{X}}$.

- In Tables 1,2,3, reporting the standard deviation on three runs is meaningless - means do not concentrate in any meaningful way with three trials. Indeed, it would be better to report the actual numbers seen instead of a misleading $\pm$ in this case. Also, the number of runs should be included in every table caption rather than just hidden away in one line away from them (it took me a good few minutes to find the number 3 here, but this is important information to contextualise the table, and should be easily available from it).

**Strength And Weaknesses:**

I think that the question of what teacher one should use for distillation is an interesting one, and would likely be of interest to the ICLR community (especially if the answer is different from 'the most accurate teacher'). However, I don't think that the authors meaningfully address this question.

I'll split this critique into two parts

Section 3: In my opinion, the theoretical study in this paper has limited novelty, and limited relevance.

For the former, I note that recovery of the regression function is well established in nonparametric statistics under smoothness assumptions as studied by this paper - for instance see Tsybakov's  work from the mid 2000s (e.g. Optimal Aggregation of Classifiers, Ann Stat 2004, and references therein). This literature establishes concrete rates for the recovery of Lipschitz regression functions (and subsequent work further studies adaptivity to this parameter, e.g. some of the work of Kpotufe and coauthors). The interested reader will also find that the (minimax optimal) rates with sample size of this nonparametric problem (without further noise assumptions) are _much_ poorer than the $1/\sqrt{n}$ that appears in the paper. What gives?

This brings me to the relevance part: in my opinion, the results shown do not actually demonstrate that $p^*$ is learnable. Indeed, the issue of precisely how $r$ affects the data is not dealt with at all. This is crucial, since $r$ appears in the bias term of the error bound (as the $Lr$ and $L^* r$ terms) - this means that if $r$ is very large, then the result is vacuous. However, if $r$ is very small, then the results is similarly vacuous due to the high scaling of $N_r$ with $r$! Indeed, observe that in high dimensions, the volume of an $r$ ball for $r \ll 1$ is extremely small. So, for instance, if the data is drawn uniformly over a unit ball in $d$-dimensions, the chance of finding even one point within a distance $r$ of a given datapoint is $O(r^{-d})$, exponentially small in the dimension. This means that for reasonably small $r$, $N^r \approx N$ so long as $N \ll r^{-d},$ and so the result is just a vacuous bound.

The above consideration must also be coupled with the fact that Neural Nets, which this theory is meant to apply to, do not have a meaningfully small Lipschitz constant (whatever the metric being used) - the results of Bartlett et al that are cited within the paper only show that the Lipschitz constant is well correlated with the generalisation error (not that the Lipschitz constant of the network is small).

Given this, I find Section 3 to both be unoriginal and irrelevant. In my opinion this is already a very serious flaw with the paper.

Section 4: Orthogonally to the above, we may ask if the empirical study of the paper is meaningful. In my opinion, this also does not pass muster. There's two aspects to this

- I think the paper does a poor job of demonstrating that the improvements in the student performance come from improvement in the fitting of $p^*$: Given the focus of the rest of the paper, the implicit statement is that the regularisation of the network allows it to learn the regression function better, and hence improves the student. However, there are obvious alternate hypotheses for this: for instance, it could be the case that the regularisation results in a smoothened teacher, and this reduction in teacher complexity makes it easier for a student to fit (or improve upon) the teacher. There are at least two tests for a hypothesis like this - first to support their claim, the authors need to run simulation studies which actually demonstrate that the teacher recovers the regression function better when regularised. Second, they should train the student with a post-hoc smoothened teacher (so train a teacher normally, and then do some local smoothing to generate soft labels) - if the improvement in student performance is similar, then the improvement cannot entirely be attributed to fitting $p^*$ better.

- The algorithmic contributions are minimal: Note that the concrete proposal of the paper is not a novel algorithm, but instead a novel application of existing methods (LR and CR) to the problem of choosing a teacher for distillation. While this is certainly interesting, I question if this is enough to merit publication in ICLR on its own strength - there are certainly a plethora of tricks associated with training DNNs for a variety of problems and domains that are not published in such a conference even though they yield improvements in the resulting models. Typically such a conference requires that the phenomenon underlying the method is well explained and clearly demonstrated, which this paper in my opinion fails to do.

**Summary Of The Paper:**

This paper is concerned with the following question: "what sort of teacher should we fit so that a resulting student trained via distillation from this teacher has good performance". Following recent work on the utility of distillation, which in particular observes that distillation benefits a student by providing a soft signal that may closely approximate the regreression function (i.e. the conditional law of the label $y$ given the input $x,$ henceforth denoted as $p^*(x)$), the paper adopts two directions: firstly, to argue that this regression function can indeed be learnt, and secondly, to investigate how one may improve the learning of the underlying law.

To this end, the authors first theoretically explore the recovery of the regression function, specifically under the assumption that $p^*$ is locally Lipschitz at a given scale, and argue that the ERM over such a class recovers the the true $p^*$ up to limited error at every datapoint in the training set. Secondly, the authors propose to use existing Lipschitz and Consistency regularised training methods to recover a teacher in a procedure they term "SoTeacher". It is empirically argued that while this results in lower teacher accuracy, the accuracy of the corresponding student model increases.

**Summary Of The Review:**

On the whole, I find this to be a rather poor submission - while the problem being considered is interesting, the method proposed is overall incremental, and the justification presented for it is entirely insufficient. I would strongly recommend rejection.

---

> ### Author Response · Authors · 2022-11-23
> **Replies to Reviewer gTte**
>
> Thanks for your valuable comments. Please check our replies below.
>
> **____**
>
> **Q. Compared to previous works on regression recovery**
>
> We wish to argue that our study is novel compared to the classic regression recovery problem. We emphasize that the goal of this paper is to learn the regression function (or true label distribution termed in our paper) **in standard teacher training practice**, rather than to learn the regression function in a general sense. This is crucial to fully understand and improve the effectiveness of knowledge distillation. The standard teacher training practice here includes two aspects.
> * First, we focus on **empirical risk minimization**. We appreciate the reviewer bringing those methods from non-parametric statistics to our attention. However, most of these methods significantly differ from empirical risk minimization. Whether these methods can learn the regression function well in practice, or have solid theoretical guarantees, cannot help understand nor improve the standard knowledge distillation practice.
>
>
> * Second, we focus on **training data reuse**, namely the teacher and student are trained on the same set of empirical data. Note that learning the regression function is technically not a problem on unseen data, even for empirical risk minimization, since it is known that the regression estimation problem can be reduced to risk minimizing on unseen data (Vapnik 1998.), as long as the loss is properly defined. However, on training data, the situation is fundamentally different as it clearly cannot be reduced to risk minimizing. Any risk minimization will trivially yield one-hot prediction for each training instance as deep neural networks can almost always fit the training data perfectly. This will contradict the empirical observation that training data reuse can significantly improve student performance, and call for the understanding of regression recovery on the training data, which we believe is not studied in previous works.
>
>
> **____**
>
> **Q. $r$ in the error bound**
>
> We agree with the reviewer that $r$ has a trade-off effect in our error bound. We expect $r$ to be relatively small, but we argue that the bound is not vacuous with a small $r$. Note that we only expect an $r$-external covering to cover the training set, instead of the entire input space. These are crucially different. First, real data does not distribute uniformly in the entire input space, but rather distribute in a small subspace, which we usually refer to as the support. Furthermore, even on the support, real data does not distribute uniformly, but is rather concentrated, possibly in a class-wise manner. These together indicate that a training set will have a smaller covering number $N_r$.

---

### Decision · Program_Chairs · 2023-01-20

**Decision:**

Reject

**Justification For Why Not Higher Score:**

Issues in the theoretical analysis: missing assumptions, and unclear proof details.

**Justification For Why Not Lower Score:**

N/A

**Metareview: Summary, Strengths And Weaknesses:**

The paper proposes to improve knowledge distillation by modifying the teacher training objective, with additional Lipschitz and consistency regularization. This is accompanied with theoretical analyses of the ability of such regularization to allow recovery of the underlying label distribution.

While reviewers found some ideas interesting, two reviewers raised concerns on the theoretical claims either lacking explicit statement of assumptions, and having some unclear proof details. From my reading, I agree these concerns are reasonable. The authors responded to some of these concerns, but the lack of a revision with the explicit changes makes it difficult to confidently affirm the satisfactory resolution of these points.

With this, we believe the paper needs more work prior to publication.

**Summary Of Ac-Reviewer Meeting:**

N/A